# A novel method for estimating properties of attentional oscillators reveals an age-related decline in flexibility

**Ece Kaya[1,2]\*, Sonja A Kotz[2,3], Molly J Henry[1,4]**

[1]Max Planck Institute for Empirical Aesthetics, Frankfurt, Germany; [2]Maastricht University, Maastricht, Netherlands; [3]Max Planck Institute for Human Cognitive and Brain Sciences, Leipzig, Germany; [4]Toronto Metropolitan University, Toronto, Canada

**\*For correspondence:**
ece.kaya@ae.mpg.de

**Competing interest:** The authors declare that no competing interests exist.

**Abstract** Dynamic attending theory proposes that the ability to track temporal cues in the auditory environment is governed by entrainment, the synchronization between internal oscillations and regularities in external auditory signals. Here, we focused on two key properties of internal oscillators: their preferred rate, the default rate in the absence of any input; and their flexibility, how they adapt to changes in rhythmic context. We developed methods to estimate oscillator properties (Experiment 1) and compared the estimates across tasks and individuals (Experiment 2). Preferred rates, estimated as the stimulus rates with peak performance, showed a harmonic relationship across measurements and were correlated with individuals' *spontaneous motor tempo*. Estimates from motor tasks were slower than those from the perceptual task, and the degree of slowing was consistent for each individual. Task performance decreased with trial-to-trial changes in stimulus rate, and responses on individual trials were biased toward the preceding trial's stimulus properties. Flexibility, quantified as an individual's ability to adapt to faster-than-previous rates, decreased with age. These findings show domain-specific rate preferences for the assumed oscillatory system underlying rhythm perception and production, and that this system loses its ability to flexibly adapt to changes in the external rhythmic context during aging.

## eLife assessment

This **important** study has practical and theoretical implications for understanding rhythm perception and production in human cognition. The evidence for individual frequency preferences and a deterioration in frequency adaptation with age is **convincing**. These findings will inform existing models of rhythm perception and production, and the reported effects of age may have clinical implications.

## Introduction

Auditory tasks such as understanding speech and listening to music rely on our ability to allocate and adjust attention to rhythmic cues in complex auditory signals. However, listeners' attention to rhythmic cues can fail when the signal is temporally disorganized (*Zalta et al., 2020*), or with advancing age (*Schneider et al., 2005*). These failures of attention might result in reduced speech comprehension (*Schneider et al., 2005*) as well as in diminished ability to solve the 'cocktail party problem' (*Zion Golumbic et al., 2013*). However, speech perception (*Poeppel and Assaneo, 2020*) and production of musical sequences are improved when stimuli are presented at specific rates (*Zamm et al., 2018*; *Scheurich et al., 2018*), indicating that these abilities might be 'restored' in certain conditions. Here, we aimed to understand factors that facilitate and impede auditory rhythm processing from two different perspectives: the factors that arise from stimulus properties in the external world and

those that stem from individual differences (the perceiver). Specifically, we tested how stimulus and the rhythmic context in which a stimulus is presented affect rhythm perception and production, and how temporal adaptation abilities change with advancing age. We found (1) a range of rates specific for each individual that yielded best performance and (2) deteriorating performance when switching between stimulus rates that was further amplified by age.

Two main theoretical approaches explain how we perceive time and rhythm. A timekeeper account proposes that the duration between two events is represented by the count of accumulated pulses that are generated by an internal pacemaker (*Scheurich et al., 2018*). An entrainment account, dynamic attending theory (DAT) proposes that biological systems consist of internal oscillations, i.e., rhythms, that adjust their phase and period to the temporal regularities of an external signal (*Jones, 2018*; *Jones, 1976*; *Jones and Boltz, 1989*). Synchronization between internal and external rhythms, termed *entrainment*, is the underlying mechanism for time and rhythm perception. Predictions of DAT have been confirmed in a number of studies that reported rhythmic facilitation effects, where a rhythmic cue improves perceptual timing of subsequent targets, with the highest accuracy for targets aligning with the entraining attentional oscillator's peaks (*Large and Jones, 1999*; *Barnes and Jones, 2000*; *Jones et al., 2002*; *McAuley and Jones, 2003*; *Martin et al., 2005*; *Herrmann et al., 2016*; *Jones et al., 2017*; *Cheng and Creel, 2020*).

The current study did not test whether timing abilities are governed by entrainment or timekeeper mechanisms. We rather adopt an entrainment approach as well as common assumptions of entrainment models (*Jones, 2008*) that derive from the general properties of limit-cycle oscillators:

> Assumption 1: Oscillators are self-sustaining; they persist even when no stimulus is present. They induce series of periodic expectations at the peaks of the oscillations.
> Assumption 2: Oscillators are adaptive; they respond to timing perturbations (e.g. changes in stimulus rate) by correcting their phase and period.
> Assumption 3: Each oscillator has an intrinsic period (*Drake et al., 2000*) at which it oscillates in the absence of any input (see Assumption 1) and is most stable against perturbations.
> Assumption 4: Oscillators can respond to stimulus rates with integer-ratio relationships (i.e. in nested hierarchies).

Two key properties of internal oscillators that were the focus of the current study are their *preferred rate* and their *flexibility*. Preferred rate, also termed as natural frequency or *eigenfrequency* in different literatures, refers to the intrinsic period of the oscillator (Assumption 3), or group of nested oscillators (*Jones, 2008*), in the absence of any input (Assumption 1). Oscillators accomplish synchronization to periodicities in the external signal better when the signal's rate is similar to the oscillator's preferred rate (or harmonics of the preferred rate; *McAuley and Jones, 2003*) than when it is dissimilar (*Notbohm et al., 2016*). The range of rates around the oscillator's preferred rate for synchronization is referred to as the *entrainment region* (*McAuley et al., 2006*). Theoretically, knowing the preferred rate of an individual's internal oscillator would allow predicting the rates at which they would most successfully interact in a real-world listening situation.

One common method to estimate the preferred rate is the spontaneous tapping task, where participants are asked to tap their finger (*McAuley et al., 2006*; *Collyer et al., 1994*; *Schwartze and Kotz, 2015*) or a drumstick (*Drake et al., 2000*), on a desk or a sensor at a 'comfortable rate'. The preferred rate estimate, spontaneous motor tempo (SMT), measured as the mean or median of the intervals between the individual taps, tends to cluster around 500–600 ms in adults (*McAuley et al., 2006*). One potential shortcoming of using SMT as a direct measure of an internal oscillator's preferred rate is that SMT reflects a 'preference' for producing periodic movements in the absence of any interaction with the environment. Although this is indeed the definition of preferred rate, a stronger test of the degree to which SMT reflects the preferred rate of an internal oscillator would be to observe successful synchronization within – but not outside of – an entrainment region. SMT does predict timing preference and performance in other tasks: participants tend to prefer stimulus rates (i.e. preferred perceptual tempo [PPT]; *McAuley et al., 2006*) closer to their SMT (*McAuley et al., 2006*), drift back to their SMT during continuation tapping in synchronization-continuation paradigms (*Zamm et al., 2018*), and over- and underproduce stimuli that are faster and slower than their SMT, respectively (*Zamm et al., 2018*; *Scheurich et al., 2018*). However, in paradigms that involve comparison of individuals' rate preferences (*McAuley et al., 2006*) and tapping performance (*Zamm et al.,*

2018; *Scheurich et al., 2020*) across stimulus rates, stimulus conditions are tailored to individuals' SMT and are low in number. This results in a resolution that is too poor to observe an entrainment region, and often confounds SMT with the global mean stimulus rate in an experiment (*Kliger Amrani and Zion Golumbic, 2020a*). We have previously proposed a synchronization-continuation paradigm where individuals' tapping behavior on a finely sampled, broad range of stimulus rates was assessed. We estimated preferred rate as the stimulus rate with minimum tapping errors during continuation tapping (*Kaya and Henry, 2022*). However, estimating preferred rates based on a tapping paradigm cannot disentangle preferred rates of an auditory oscillator, a motor oscillator, or a coupled oscillatory system whose preferred rate would be influenced by the preferences and coupling strengths of its components (*Schneider et al., 2005*). Thus, here we applied the fine rate sampling to a perceptual paradigm (Experiment 1), estimated preferred rates in perceptual and motor versions of the paradigm with same stimulus rate conditions (Experiment 2), and compared the estimates to individuals' SMT and PPT (Experiment 2).

Based on Assumption 2, we defined flexibility as the internal oscillator's ability to adapt to rate changes in the external sound signal (*Kaya and Henry, 2022*). The logic is as follows: upon encountering a new rate, the oscillator gradually updates its phase and period to each upcoming interval. From a dynamical systems perspective, flexibility can be conceptualized as a complement to 'stiffness', and might be quantified based on the presence of hysteresis, which refers to a system's tendency to stay in a previous state despite changes in stimulus parameters (*Kelso, 1995*). An inflexible oscillator would exhibit hysteresis and continue to respond in a way that reflects the properties of previously entrained stimuli. A fully flexible oscillator would not exhibit hysteresis as it would completely update its phase and period to the new stimulus, resulting in no discrepancy between the current stimulus and its internal representation. Thus, the extent to which timing performance would be affected by the stimulus history is inversely related to the underlying oscillator's flexibility.

Prior research reveals effects of preceding context, also referred to as serial dependence (*Kim and Alais, 2021*; *Motala et al., 2020*) and carryover effects (*Wiener et al., 2014*), on timing behavior in tasks with and without a motor synchronization component. Within individual trials of synchronized tapping paradigms, changes in stimulus rate (period perturbation) and stimulus onset times (phase perturbation) result in increased asynchronies between stimulus and tap onsets. This effect is more pronounced for phase than period perturbations (*Large et al., 2002*; *Loehr et al., 2011*), and for sequences that speed up than those that slow down (*Scheurich et al., 2020*; *Loehr et al., 2011*). Across trials, the tapping rate in each trial is biased toward the previous trial's stimulus rate (*Kaya and Henry, 2022*; *Motala et al., 2020*). Temporal judgments in the absence of motor synchronization are also affected by the stimulus properties presented in a preceding trial (*Wiener et al., 2014*; *Jones and Mcauley, 2005*; *Wiener and Thompson, 2015*) and throughout the experiment (*Jones and Mcauley, 2005*; *McAuley and Miller, 2007*), suggesting effects of local and global temporal contexts on duration perception. The majority of studies that revealed individual differences in proneness to history effects (*Kim and Alais, 2021*; *Arzounian et al., 2017*) have not aimed to explicitly estimate the extent and source of these individual differences, or have done so in shorter temporal contexts, using different operational definitions of flexibility than the one used here (*Scheurich et al., 2018*). Finally, similar to methods proposed to estimate preferred rate (*Zamm et al., 2018*; *Scheurich et al., 2018*; *McAuley et al., 2006*; *Kaya and Henry, 2022*; *McPherson et al., 2018*), previous attempts to measure flexibility (*Scheurich et al., 2018*; *Scheurich et al., 2020*; *Kaya and Henry, 2022*) involved only motor responses. Thus, we presented the same stimulus history to participants in two tasks, one with and one without the motor demands of synchronize-continue tapping. This design allowed assessing the effects of the same predictor (trial-to-trial rate change) on performance in different tasks, and thereby performing systematic comparisons of oscillator flexibility across perceptual and motor domains.

From the perceiver's side, we chose to focus on how properties of internal oscillators change with advancing age. Studies assessing age-related changes in timing abilities show that older, as compared to younger individuals, produce slower tapping rates when asked to tap at a comfortable rate (*McAuley et al., 2006*; *Baudouin et al., 2004*) and at the fastest rate (*Turgeon et al., 2011*) they can maintain, show worse performance in temporal-order judgments (*Szymaszek et al., 2009*), gap detection (*Fitzgibbons and Gordon-Salant, 1995*) and discrimination and reproduction of time intervals (*Incao et al., 2022*), and tend to prefer slower stimulus rates (*McAuley et al., 2006*), which

manifests in a breakdown in understanding fast speech. From an entrainment perspective, these findings suggest that internal oscillators of older individuals have slower preferred rates, reduced flexibility, or both. While the current study did not incorporate neural measures, it is worth noting that literature on neural entrainment can offer insights into the dynamics of attention. This is particularly relevant as these physical measures often align with the predictions of DAT (see *Haegens and Zion Golumbic, 2018*; *Henry and Herrmann, 2014* for reviews). Neural entrainment to external auditory signals is aberrant (*Goossens et al., 2016*; *Herrmann et al., 2019*; *Purcell et al., 2004*), and less responsive to top-down attention in older than younger adults (*Henry et al., 2017*). Moreover, older adults exhibit reduced neural adaptation (*Herrmann et al., 2023*) and sensory gating (*Brinkmann et al., 2021*), suggesting an age-related decline in neural inhibition (*Herrmann et al., 2023*) that leads to a reduced capacity of the auditory system to adapt based on context. Based on the behavioral findings converging on reduced temporal abilities and evidence for impaired neural entrainment in older individuals, we hypothesized that older adults would exhibit stronger hysteresis than younger adults, which should result in smaller estimates of oscillator flexibility.

The aim of the current study was to estimate individuals' preferred rate and flexibility in rhythmic tasks with and without a motor synchronization component, and in both preference and performance contexts: here, preference refers to SMT and PPT, whereas performance refers to tasks that require listeners to either synchronize with or make a perceptual judgment about rhythmic stimuli. Moreover, we aimed to assess how internal oscillator properties, specifically oscillator flexibility, change with advancing age.

We conducted two experiments. The main goal of Experiment 1 was to develop methods to estimate preferred rate and flexibility in a paradigm without a motor synchronization component, as a complement to our recent tapping study (*Kaya and Henry, 2022*). The task was a duration discrimination paradigm where participants compared the duration of a single comparison interval to the duration of intervals making up a standard stimulus. We assessed the effect of stimulus history on responses by comparing performance across two sessions with the same finely sampled pool of stimulus rates, one where we maximized and the other where we minimized the amount of rate change across trials. Experiment 2 involved shorter versions of the duration discrimination (Experiment 1) and paced tapping (*Kaya and Henry, 2022*) tasks with matched stimulus rates and histories, unpaced tapping tasks including SMT, and two tasks where individuals' rate preferences (PPT) were measured.

In line with the *preferred period hypothesis* (*McAuley et al., 2006*), if SMT captures the preferred rate of common mechanisms underlying rhythm perception and production, we should see better performance around an individual's SMT, as has previously been observed for motor tasks (*Zamm et al., 2018*; *Scheurich et al., 2018*; *McAuley et al., 2006*; *Kliger Amrani and Zion Golumbic, 2020b*). However, we did not necessarily expect a one-to-one correspondence between preferred rate estimates across tasks with and without a motor component, as individual differences in motor contributions to synchronization abilities are well documented (*Assaneo et al., 2021*).

We hypothesized that larger trial-to-trial changes in stimulus rate would lead to poorer performance due to hysteresis, in that both tapping and duration discrimination responses should reflect the properties of the preceding stimuli. Thus, we expected that larger changes between consecutive trials' stimulus rates should decrease discrimination accuracy and increase tapping errors. We expected that the strength of these effects – the degree of inflexibility – should increase with age.

# Experiment 1
## Methods
### Participants

Participants (N=31) were recruited from the participant pool of Max Planck Institute for Empirical Aesthetics laboratories in Frankfurt, Germany. Written informed consent was obtained from all participants. The procedure was approved by the Ethics Council of the Max Planck Society (approval number 2019_04) and the Research Ethics Board at Toronto Metropolitan University in accordance with the Declaration of Helsinki. Out of 31 (age: M=33, SD = 11) individuals who were recruited for the study, 27 participants (age: M=33, SD = 12) completed both sessions. Upon completion of each session, participants received 7 euros for every 30 min of their participation (21 euros per session on average). Two participants volunteered to complete the study without compensation. Prior to the experimental

**Figure 1.** Design of the duration discrimination task in Experiment 1. Each trial consisted of an isochronous standard sequence of five sounds (four intervals), followed by silence and another pair of sounds. The comparison duration was either shorter or longer than the standard intervals and took on one of ten values (DEV) that were proportional to the inter-onset interval (IOI) between sounds making up the standard sequence. The task was to press the S or L key to indicate whether the comparison interval was shorter or longer than the standard IOI. Over the course of 400 unique trials of a single session, IOI ranged from 200 ms to 998 ms. In random-order sessions, change in stimulus rate between a given trial n and immediately preceding trial n–1 ($\Delta$IOI) was maximized, and the distribution of $\Delta$IOI ranged from –778 ms to +770 ms. In linear-order sessions, IOI increased in each trial in the first 200 trials and decreased in the other half of the trials (or vice versa, counterbalanced across participants) in steps of 4 ms.

sessions, participants completed an online survey. All participants self-reported normal hearing and proficiency in English.

## Procedure

The study consisted of an online background survey that participants completed at home, and then two experimental sessions. During the in-lab experimental sessions, participants completed two types of tasks. A series of unpaced tapping tasks, consisting of SMT and a 'forced' motor tempo (FMT) task, which was used to assess the range of free tapping rates within the participants' motor abilities; and the main task, duration discrimination, where participants judged whether a comparison interval was 'shorter' or 'longer' than the intervals making up a standard sequence. Details of all tasks are provided below. Sessions were separated by 4–19 days. A single session started with the SMT and FMT tasks. Participants then set the sound volume to a level that they found comfortable for completing the task. Then, participants were presented with instructions on a computer screen that explained the main task with text and figures. A practice block, simulating the duration discrimination task, followed the instructions (details below). All instructions were in English. Once participants indicated that they understood the task, the main task blocks were initiated. Finally, unpaced tapping tasks were repeated in the same order. Participants were debriefed upon their request, only after the second session. An individual session lasted 90 min on average.

## Duration discrimination task

The main task was a duration discrimination paradigm, where participants judged whether a comparison interval was longer or shorter than the intervals making up an isochronous standard sequence, by pressing either the L (longer) or S (shorter) key on a computer keyboard. The task procedure is illustrated in *Figure 1*. In each experimental session, 400 unique trials of this task were presented, each consisting of a combination of the three main independent variables: the inter-onset interval, IOI; amount of deviation of the comparison interval from the standard, DEV, and the amount of change in stimulus IOI between consecutive trials, $\Delta$IOI. We explain each of these variables in detail in the next paragraphs.

Stimuli were made up of 50 ms woodblock sounds; first, an isochronous standard sequence and then a comparison interval, separated by a silent gap. The interval between the five woodblock sounds making up the 'standard' isochronous stimulus sequence is referred to as IOI. Each trial's IOI was drawn (without replacement) from a pool of all possible stimulus rates, linearly spaced between 200 ms to 998 ms in 2 ms steps. The silent interval between the last stimulus onset of the standard sequence and the first stimulus onset of the comparison pair was six times the standard IOI.

The comparison interval on each trial was longer or shorter than the standard IOI. DEV refers to the magnitude of the comparison interval's deviation from the standard IOI. DEV took on one of ten levels, which were proportional to IOI:±2%, 7%, 11%, 16%, 20%. Each DEV level was presented 40 times in each session. Since IOI was unique on each trial, IOI and DEV were not fully crossed factors.

Instead, the IOI dimension was divided into 40 bins, each consisting of 10 consecutive IOIs. The 10 DEV levels were randomly assigned to the 10 IOI values in each bin. The correspondence between IOI and DEV pairs was unique for each participant.

While the mean (M=599 ms), standard deviation (SD = 231 ms), and range (200 ms, 998 ms) of the presented stimulus IOIs were identical between the sessions, the way IOI changed from trial to trial was different. Change in IOI between consecutive trials was referred to as ΔIOI. In one session, the 'linear-order' session, ΔIOI was always ±4 ms. In one half of the session, ΔIOI was fixed at +4 ms. That is, IOI was 200 ms in the first trial, 204 ms in the second, and so on. In the other half of the session, ΔIOI was fixed at –4 ms. On the first trial, IOI was 998 ms, 994 ms in the second, and so on. The starting point, either 200 ms or 998 ms (in fast-start and slow-start conditions, respectively), was counterbalanced across participants.

In the other session, the 'random-order' session, ΔIOI was maximized, and the direction of the change (i.e. whether a trial was faster or slower than the previous) alternated on every trial. That is, if the stimulus IOI on one trial was faster than the previous (–ΔIOI), it would be slower (+ΔIOI) in the following trial, and vice versa. Note that stimulus IOI was stable within the standard sequence, and only changed between trials. Session order, i.e., whether a participant experienced the linear-order or random-order session first, was counterbalanced across participants. An example trajectory of stimulus IOI within random-order and linear-order sessions across trials is illustrated in *Figure 1*.

In each session, participants completed 407 trials, presented in 8 blocks with 50 trials in the first block, and 51 trials in the remaining 7 blocks. Except for the first block, the first trial of each block repeated the IOI that was presented as the last trial of the preceding block and was discarded from further analyses; this enabled preservation of the between-trial histories across blocks between which participants were allowed to take short breaks. Before the main task, participants were instructed about the task, and practiced the task for at least 6 trials. Instructions included two example trials with IOI of 500 ms, one with DEV of +0.3 and another with DEV of –0.3, illustrating 'comparison longer' and 'comparison shorter' conditions, respectively. DEV was fixed at +0.2 in half of the practice trials and at –0.2 in the other half. Two practice trials each were presented at fast, medium, and slow IOIs; randomly selected from ranges of [300–500 ms], [501–700 ms], and [701–900 ms], respectively. If participants failed on more than 3 of the first 6 practice trials, they completed another round of 6 practice trials. Both example and practice trials were randomly ordered within their respective blocks in each session.

The dependent variables were accuracy and bias. Accuracy coded whether a response on a trial was correct or not (1=correct, 0=incorrect). Bias, on the other hand, could take on one of three values per trial: if the response was correct, bias was 0. If the comparison interval in a trial was longer than the standard, and the participant's response was 'shorter', bias in that trial was –1. Similarly, if participant's response was 'longer' in a trial where comparison interval was shorter, bias was +1.

## Unpaced tapping tasks

Unpaced tapping tasks consisted of a single SMT trial and two FMT trials, one each to estimate the 'slowest' and 'fastest' rates at which participants could maintain steady tapping. The unpaced tasks were repeated in the same order before and after completion of the duration discrimination task in both sessions. In the SMT task, participants were instructed to 'tap on the desk at a rate that is comfortable to maintain'. In the FMT tasks, the instruction was 'tap at the slowest rate that is comfortable to maintain' (FMT-slowest) and to 'tap at the fastest rate that is comfortable to maintain' (FMT-fastest). Participants tapped for 30 s in the SMT task and FMT-fastest task, and 45 s in the FMT-slowest task. For all unpaced tapping tasks, the dependent measures were tapping rate (median of the produced intervals) and coefficient of variation (CV).

## Apparatus

Stimuli were generated and presented on a Windows desktop computer, using the Psychophysics Toolbox extensions (*Brainard, 1997*; *Pelli, 1997*) for MATLAB. Auditory stimuli were presented via Beyerdynamics 880 Pro headphones. The audio signal was presented and recorded by an RME Fireface UC soundcard. All instructions were presented on an ASUS VG24QE LCD screen. Keypress responses for the duration discrimination task were collected on a USB keyboard. Tapping responses for the unpaced tapping tasks were recorded via a Schaller Oyster S/P contact microphone at a sampling rate

of 44,100 Hz. The contact microphone was attached on the right half of the desk by default. Prior to the sessions, participants were asked to specify if they would like the microphone to be moved to the left half of the desk. None of the participants requested a relocation of the microphone.

## Background survey

Prior to the first experimental session, participants completed an online survey. The survey consisted of two parts: the first part included questions about participants' demographics, language skills, hearing abilities, and psychological disorders. The second part was 'The Goldsmiths Musical Sophistication Index', 'Gold-MSI' (*Müllensiefen et al., 2014*). The survey language was English by default, with an option to change the language to German. One question in the Gold-MSI was removed from the analyses due to contrasting Likert coding between the different languages in which the survey was completed.

## Analysis

### Data cleaning and exclusion criteria

The raw format of the tapping data was audio, since tapping responses were collected by a microphone. Individual taps were extracted from the audio files after visual inspection of the soundwave of each trial to set the noise floor for the recording on that trial. All peaks that exceeded the noise floor were retained. Inter-tap intervals (ITIs) were calculated as the difference between neighboring taps' timestamps. We developed an automated procedure that detects and removes single-trial ITI outliers while accounting for drift that may have occurred within tapping trials. The script first marked the ITIs whose deviation from the median ITI exceeded 3× the median absolute deviation (MAD) of all ITIs in the respective trial. Then, it fitted a linear regression to the unmarked ITIs as a function of tap count. Finally, it removed any ITI that was smaller than half or larger than 1.5 times the predicted ITI.

Exclusion criteria for the main task were (1) a decrease in accuracy with increasing absolute DEV, and (2) chance level performance for both deviation directions (trials where comparison interval was shorter, and those where it was longer). To assess the first criterion at the participant level, we fitted separate models to each individual's single-session data where accuracy was predicted by absolute deviation of the comparison interval for either shorter ($|{-}DEV|$) or longer ($|{+}DEV|$) comparison conditions. The models were fitted using MATLAB's *fitglm* function, with the response variable distribution specified as 'binomial', and link function specified as 'logit', since the response variable, accuracy, was binary. Next, we compared the slopes (β) obtained from the separate models where either $|{-}DEV|$ or $|{+}DEV|$ predicted accuracy against zero, using one-tailed one-sample t-tests. All participants had positive slopes for both directions in both session types, indicating that the probability of correct response increased with $|DEV|$ in all conditions. To test for chance level performance, for each session type, we split all trials into negative and positive DEV conditions and compared each group of trials' accuracy against a mean of 0.5, using one-sample t-tests. Results showed that none of the participants had chance-level performance for both deviation directions. Finally, before applying group-level statistics such as t-tests and correlations, any data point that fell outside of the interquartile range was excluded from the respective distributions.

### Preferred rate estimates

We conceptualized individuals' preferred rates as the stimulus rates where duration discrimination accuracy was highest. To estimate preferred rate on an individual basis, we smoothed response accuracy across the stimulus rate (IOI) dimension for each session type, using the *smoothdata* function in MATLAB, which outputs the moving average of the neighboring data points within a specified window size. We used 'Gaussian' as the method for smoothing that calculates the Gaussian-weighted moving average over each window. This method gives higher values into the midpoint of the window, enhancing the fluctuations in the data that were the focus of the current analysis. As we were interested in a single-point maximum accuracy for each individual and session, we optimized the window size for each session type such that the smoothed data revealed a single global maximum. An illustration of the optimization for an example participant's dataset is shown in *Figure 2—figure supplement 1*. For small windows, smoothed data included multiple IOI values where accuracy was 1, especially in the linear-order sessions. The optimization procedure revealed that, to obtain a single global maximum for each individual's dataset, accuracy should be smoothed by windows of 26 samples in

the random-order sessions and 48 samples in linear-order sessions (*Figure 2—figure supplement 1*). To equalize the smoothing across the variables of accuracy and IOI, we also smoothed IOI with the same window size. Estimates of preferred rate were taken as the smoothed IOI that yielded maximum accuracy.

To compare the preferred rate estimates between session types, we first conducted a paired-samples t-test. Then, we assessed the correspondence between the estimates. However, conventional correlation methods are not able to capture possible harmonic relationships between variables. Thus, we used a permutation test that accounted for the harmonic structure in data, in addition to the assessment of one-to-one correspondence between the data points. The test first calculates the perpendicular distance of the data points to the closest line among the y=x, y=2*x, and y=x/2 theoretical lines (referred to as residuals here, as in *Kaya and Henry, 2022*) whose sum quantifies how much the data points deviate from a total harmonic correspondence. Then, the test shuffles the Y axis values with respect to the X axis values 1000 times and calculates summed residuals for each permutation. The p-value is the percentage of summed residuals smaller than the initial value computed from original data. To validate the results obtained from this test, we ran an additional analysis using a modular approach. We first calculated how much the slower estimate (larger IOI value) diverts, proportionally from the faster estimate (smaller IOI value) or its multiples (i.e. harmonics) by normalizing the estimates from both sessions by the faster estimate. The outcome measure was the modulus of the slower, with respect to the faster estimate, divided by the faster estimate, described as mod(max(X), min(X))/min(X) where X = [session1_estimate session2_estimate]. For example, if a participant's preferred rate estimate is 603 ms in one session, 295 ms in the other session, the slower estimate (603 ms) diverts from the *multiple* of the faster estimate (590 ms) by 13 ms, a proportional deviation of 4% of the faster estimate. As the resulting distribution of percentage diversion values was non-normal, we used median to summarize the central tendency for percentage diversion of slow from fast preferred rate estimates. Then, we ran a permutation test where linear-order session estimates were shuffled over 1000 iterations, and median percentage diversion values for each iteration (*Figure 2—figure supplement 2*) were retrieved. This test statistic was significant (p=0.003), indicating that the harmonic relationships we observed in the estimates were not due to chance or dependent on the assessment method.

In addition to estimating preferred rate at stimulus rates with peak performance, we investigated whether accuracy increased as a function of detuning, namely, the difference between stimulus rate and preferred rate, as predicted by the entrainment models (*Jones, 2018*; *Large, 1994*; *McAuley, 1995*). We tested this prediction by assessing the slopes of mixed-effects logistic regression models, where accuracy was regressed on the IOI condition, separately for stimulus rates that were faster or slower than an individual's preferred rate estimate. To do so, we first z-scored IOIs that were faster and slower than the participant's preferred rate estimates separately to render IOI scales comparable across participants. The detuning direction (i.e. whether stimulus IOI was faster or slower than the preferred rate estimate) was coded categorically. Accuracy (binary) was predicted by these variables (z-scored IOI, detuning direction), and their interaction. The model was fitted separately to datasets from random-order and linear-order sessions, using the *fitglme* function in MATLAB. Fixed effects were z-scored IOI and detuning direction and random effect was their interaction. We expected a systematic increase in performance toward the preferred rate, which would result in a significant interaction between stimulus rate and detuning direction. To decompose the significant interaction and to visualize the effects of detuning, we fitted separate models to each participant's single-session datasets, and obtained slopes from each direction condition, hereafter denoted as the 'relative-detuning slope'. We treated relative-detuning slope as an index of the magnitude of relative-detuning effects on accuracy. We then evaluated these models, using the *glmval* function in MATLAB to obtain *predicted* accuracy values for each participant and session. To visualize the relative-detuning curves, we averaged the predicted accuracies across participants within each session, separately for each direction condition (faster or slower than the preferred rate). To obtain a single value of relative-detuning magnitude for each participant, we averaged relative-detuning slopes across direction conditions. However, since slopes from IOI > preferred rate conditions quantified an accuracy decrease as a function of detuning, we sign-flipped these slopes before averaging. The resulting average relative-detuning slopes, obtained from each participant's single-session datasets, quantified how much the accuracy increase toward preferred rate was dependent on, in other words, sensitive to, relative detuning.

## Flexibility estimates

We hypothesized that larger trial-to-trial changes in stimulus rate would reduce accuracy. To test this hypothesis, we first compared participants' average accuracy between session types, using a paired-sample t-test. Then, we assessed the effect of absolute rate change (|±ΔIOI|) on accuracy for each individual. To do so, we fitted generalized linear models to each participant's random-order session data and obtained slopes (β) that quantified the strength of the |±ΔIOI| effect for each participant. The models were fitted using MATLAB's *fitglm* function, with the distribution of the response variable specified as 'binomial', and link function specified as 'logit', since the response variable, accuracy, was binary. We fitted separate models for trials where the stimulus was faster or slower than the previous trial's stimulus, where the predictor was either |−ΔIOI| or |+ΔIOI|, respectively. The model formula was $p(Y=1|X)=e^{(\alpha+\beta x)}/e^{(\alpha+\beta x)}+1$, where Y is accuracy and X is the amount of rate change in trials that were faster than previous (|−ΔIOI|) or in trials that were slower (|+ΔIOI|). Next, using one-tailed one-sample t-tests, we tested whether models' β were smaller than zero, which would confirm a decrease in accuracy as a function of |−ΔIOI| or |+ΔIOI|. The β values, which quantified individuals' ability to adapt to changes in stimulus rate from one trial to the next, served as our single-individual estimate of oscillator flexibility. Finally, to investigate whether responses were affected by the previous trial's stimulus, we computed participants' average bias in trials where stimulus was faster than the previous one (|−ΔIOI|), and in trials where it was slower (|+ΔIOI|). We compared the distribution of average bias values against zero, using one-sample t-tests. Non-zero positive bias indicated that participants incorrectly responded as 'comparison interval was longer' in trials where comparison interval was in fact shorter than the standard interval, and non-zero negative bias indicated the opposite. We further tested the relationship between the flexibility estimates (β from models where |−ΔIOI| or |+ΔIOI| predicted accuracy) and average relative-detuning slopes (see *Preferred rate estimates*) from random-order sessions. We predicted that flexible oscillators (larger β) would be less severely affected by detuning, and thus have smaller detuning slopes. Conversely, inflexible oscillators (smaller β) should have more difficulty in adapting to a large range of stimulus rates, and their adaptive abilities should be constrained around the preferred rate, as indexed by steeper relative-detuning slopes.

## Results

We first assessed whether accuracy increased with increasing DEV. Comparison of the distribution of slopes (β) against zero showed that for both DEV directions, β were greater than zero. Descriptive and inferential statistics are shown in ***Supplementary file 1a***. Next, we compared participants' average accuracies from 'comparison shorter' (|−DEV|) and 'comparison longer' (|+DEV|) conditions. Although average accuracy from the latter conditions was higher in both sessions, these differences were nonsignificant.

## Preferred rate estimates

We expected that accuracy should depend on IOI differently for each participant, and estimated individuals' preferred rate as the IOI where smoothed accuracy was maximum. Between-session comparisons showed that estimates did not significantly differ between sessions (p=0.129). When we directly compared preferred rate estimates from the two session types (***Figure 2A***), we found that for most participants, the estimates were numerically close to each other. Interestingly, for some participants, estimates from one session were close to double or half of those from the other session, suggesting a harmonic relationship between the estimates. We applied a permutation test that accounted for the harmonic structure of the data and found a significant relationship between estimates from two session types (p=0.008, ***Figure 2A***).

Logistic models assessing a systematic increase in accuracy toward the preferred rate estimate in each session type revealed significant main effects of IOI (linear-order session: β=0.26399, p=4.9546e-09; random-order session: β=0.17506, p=8.1406e-08), and significant interactions between IOI and direction (linear-order session: β=−0.44378, p=4.1998e-13; random-order session: β=−0.36437, p=5.0164e-15), indicating that accuracy increased as fast rates slowed toward the preferred rate (positive slopes) and decreased again as slow rates slowed further past the preferred rate (negative slopes), regardless of the session type. ***Figure 2B*** illustrates the preferred rate estimation method for an example participant's dataset and shows the predicted accuracy values from models fitted to each participant's single-session datasets. Note that the main effect and interaction

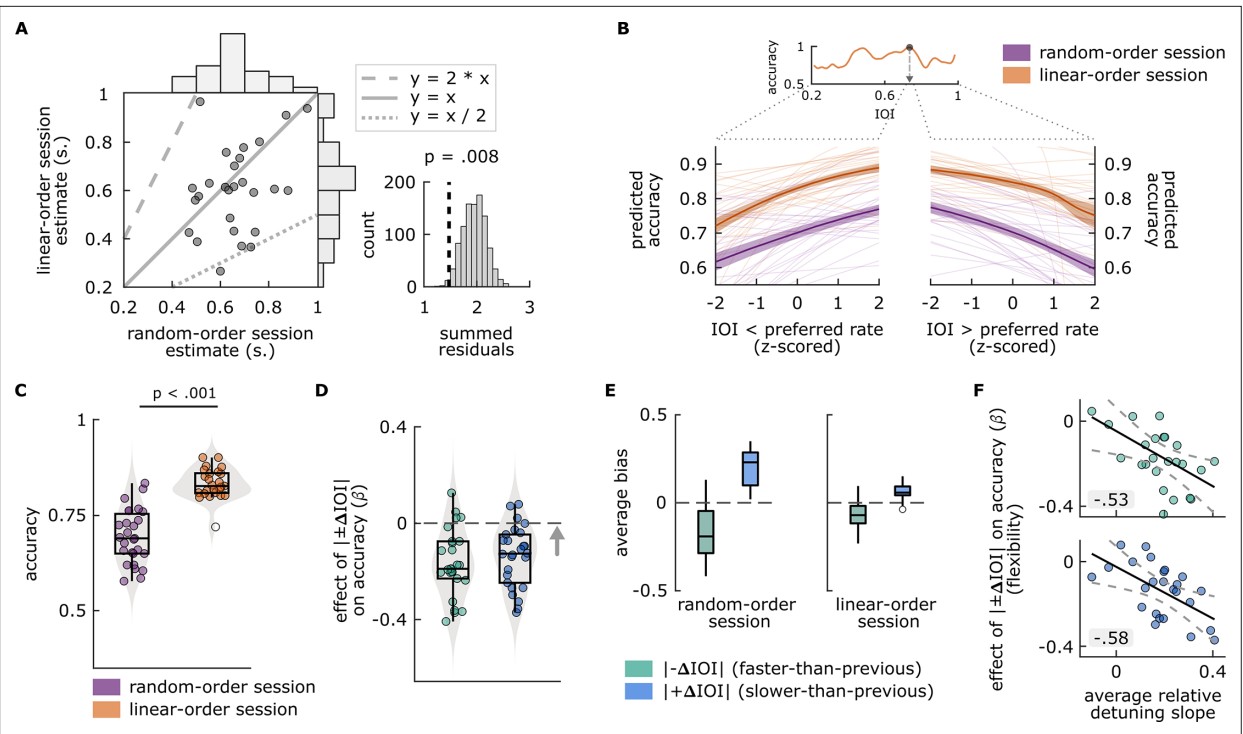

**Figure 2.** Main findings of Experiment 1. (**A**) Left: Each circle represents a single participant's preferred rate estimate from the random-order session (x axis) and linear-order session (y axis). The histograms along the top and right of the plot show the distributions of estimates for each session type. The dotted and dashed lines respectively represent 1:2 and 2:1 ratio between the axes, and the solid line represents one-to-one correspondence. Right: Permutation test results. The distribution of summed residuals (distance of data points to the closest y=x, y=2*x, and y=x/2 lines) of shuffled data over 1000 iterations, and the summed residual from original data (dashed line) that fell below 0.008 of the permutation distribution. (**B**) Top: Illustration of the preferred rate estimation method from an example participant's linear-order session dataset. Estimates were the stimulus rates (IOI) where smoothed accuracy (orange line) was maximum (arrow). The dotted lines originating from the IOI axis delineate the stimulus rates that were faster (left, IOI < preferred rate) and slower (right, IOI > preferred rate) than the preferred rate estimate and expand those separate axes, the values of which were z-scored for the relative-detuning analysis. Bottom: Predicted accuracy, calculated from single-participant models where accuracy in random-order (purple) and linear-order (orange) sessions was predicted by z-scored IOIs that were faster than a participant's preferred rate estimate (left), and by those that were slower (right). Thin lines show predicted accuracy from single-participant models, solid lines show the averages across participants, and the shaded areas represent standard error of the mean. Predicted accuracy is maximal at the preferred rate and decreases as a function of detuning. (**C**) Average accuracy from random-order (left, purple) and linear-order (right, orange) sessions. Each circle represents a participant's average accuracy. (**D**) Flexibility estimates. Each circle represents an individuals' slope (β) obtained from logistic models, fitted separately to conditions where |−ΔIOI| (left, green) or |+ΔIOI| (right blue) predicted accuracy, with greater values (arrow's direction) indicating better oscillator flexibility. The means of the distributions of β from both conditions were smaller than zero (dashed line), indicating a negative effect of between-trial absolute rate change on accuracy. (**E**) Participants' average bias from |−ΔIOI| (green) and |+ΔIOI| (blue) conditions in random-order (left) and linear-order (right) sessions. Negative bias indicates underestimation of the comparison intervals, positive bias indicates the opposite. Box plots in **C–E** show median (black vertical line), 25th and 75th percentiles (box edges), and extreme data points (whiskers). In **C** and **E**, empty circles show outlier values that remained after data cleaning procedures. (**F**) Correlations between participants' average relative-detuning slopes, indexing the steepness of the increase in accuracy toward the preferred rate estimate (from panel **B**), and flexibility estimates from |−ΔIOI| (top, green) and |+ΔIOI| (bottom, blue) conditions (from panel **C**). Solid black lines represent the best-fit line, dashed lines represent 95% confidence intervals.

The online version of this article includes the following figure supplement(s) for figure 2:

**Figure supplement 1.** Illustration of the optimization procedure and parameter choices for smoothing accuracy in Experiment 1.

**Figure supplement 2.** Permutation test results from the modular approach.

were obtained from mixed-effects models that included aggregated datasets from all participants, whereas the slopes quantifying the accuracy increase as a function of detuning (i.e. relative-detuning slopes) were from models fitted to single-participant datasets.

### Flexibility estimates

Average accuracy (*Figure 2C*) was higher in linear-order (M=0.834, SD = 0.039) sessions than in random-order (M=0.695, SD = 0.072) sessions (t(24) = 12.5964, p=4.5497e-12). β from models where |±ΔIOI| predicted accuracy was significantly smaller than zero for both |–ΔIOI| and |+ΔIOI| conditions and we found no significant differences between β from the former and latter conditions, showing that the probability of giving a correct response decreased with the amount of rate change across trials, regardless of whether a stimulus was faster or slower than the previous trial. Descriptive and inferential statistics are provided in *Supplementary file 1a*. The distributions of β from individual fits are shown in *Figure 2D*. To investigate the source of the negative relationship between |±ΔIOI| and accuracy, we analyzed how rate change affected bias. In both session types, participants' average bias from faster-than-previous (|–ΔIOI|) conditions was significantly smaller than zero (random-order session: M=–0.179, SD = 0.144, t(26) = –6.4487, p=3.9085e-07; linear-order session: M=–0.065, SD = 0.078, t(26) = –4.3159, p=0.00010215), and average bias from slower-than-previous (|+ΔIOI|) conditions was significantly greater than zero (random-order session: M=0.195, SD = 0.096, t(26) = 10.5406, p=3.5025e-11; linear-order session: M=0.063, SD = 0.046, t(23) = 6.6472, p=4.4044e-07), as shown in *Figure 2E*. These results indicate that participants perceived longer comparison intervals as shorter on the trials where stimulus was faster than the previous trial, and vice versa on trials where stimulus was slower.

We tested the relationship between the flexibility estimates and single-participant relative-detuning slopes from random-order sessions (*Figure 2B*). The results revealed negative correlations between the relative-detuning slopes and flexibility estimates, both with β(r23) = –0.52905, p=0.0065428) from models where |–ΔIOI| predicted accuracy (adapting to speeding-up trials), and β (r(23) = –0.57999, p=0.0023735) from models where |+ΔIOI| predicted accuracy (adapting to slowing-down trials). That is, the performance of individuals with less flexible oscillators suffered more as detuning increased. These results are shown in *Figure 2F*.

### Unpaced tapping

Individuals completed a series of unpaced tapping tasks in the beginning and in the end of each session. Here, we focused on tapping rate from the SMT task. We first compared individuals' SMT before and after sessions. For both random- and linear-order sessions, SMT from before and after the session correlated and were not significantly different. Given the consistency of the measure, we averaged participants' SMT within sessions and compared the mean SMT across session types. We found a strong correlation between tapping rates from the random- and linear-order sessions. Test results of the unpaced tapping analyses are provided in *Supplementary file 1b*.

## Discussion

The results of Experiment 1 showed that discrimination accuracy systematically increased with the difference between standard and comparison intervals (DEV) and decreased with the difference in stimulus rate between consecutive trials (|±ΔIOI|). Accuracy showed a nonlinear relationship with IOI: we observed improved accuracy at an individual-specific range of stimulus rates and in cases at their (sub)harmonics.

For most participants, estimates from random-order sessions were close to double the estimates from the linear-order sessions (see *Figure 2A*). Correspondence between estimates from the two session types shows the reliability of the paradigm and robustness of the methods we developed for the preferred rate estimation, since we were able to obtain similar estimates in repeated measurements, and under conditions with major differences in stimulus history and task difficulty. The current findings support three key predictions of the entrainment account. First, similar estimates of preferred rate under different temporal contexts and repeated measurements as well as a systematic increase in accuracy toward the preferred rate suggest improved timing abilities in situations with smaller detuning between the oscillator's preferred rate and the stimulus rate (*Notbohm et al., 2016*). Second, that the estimates from the more challenging random-order session were narrower while preserving

the correspondence to those from other conditions indicates that the internal oscillators were able to adaptively (*McAuley and Jones, 2003*; *McAuley, 1995*) entrain to the range of rates around their preferred rate, i.e., their entrainment region (*McAuley et al., 2006*). Finally, the harmonic relationship between the estimates from the two session types suggest the oscillator's ability to respond to multiple, nested rates, either due to the circular nature of oscillators (*McAuley, 1995*) or by involvement of multiple nested oscillators in rhythmic entrainment (*Jones, 2008*).

Two sets of results confirmed the presence of history effects on timing performance. Accuracy was lower in random-order sessions where absolute rate change ($|\pm\Delta IOI|$) was maximum, than in linear-order sessions where it was minimum. Moreover, accuracy in random-order sessions decreased as rate change increased. The difference in discrimination accuracy between sessions cannot be attributed merely to the effects of the global context, given that the global context was identical across session types. If the duration representations were drawn toward the mean of the rates presented in the session ('the central tendency effect', *Jazayeri and Shadlen, 2010*), accuracy would be similar between the sessions with identical global means. Instead, we observed a drastic decrease in accuracy in the random-order session, which suggests a stronger influence of local than global context in the current paradigm. The analyses of bias confirmed this explanation by showing that internal duration representations on a given trial were biased toward the previous stimulus rate. Interestingly, rate change across trials affected bias even when it was small and fixed.

## Experiment 2
### Methods
#### Participants
32 participants were recruited from the participant pool of Max Planck Institute for Empirical Aesthetics laboratories. The procedure was approved by the Ethics Council of the Max Planck Society (approval number 2019_04) and the Research Ethics Board at Toronto Metropolitan University and was in accordance with the Declaration of Helsinki. Participants signed an informed consent prior to the session and received 21 euros on average as compensation after completing the session. Prior to the experimental sessions, they also completed an online survey. We targeted a uniform age distribution (M=50, SD = 17): within the range of 20–80 years of age, we recruited 5 or 6 participants from each 10-year age bin.

#### Procedure
The study consisted of an online background survey, a series of unpaced tapping tasks including the SMT, two PPT tasks, a duration discrimination and a paced tapping task. Participants' hearing thresholds were measured using standard pure-tone audiometry. Participants were not excluded based on hearing threshold. The experiment procedure is illustrated in *Figure 3A*. Details of all tasks are provided below.

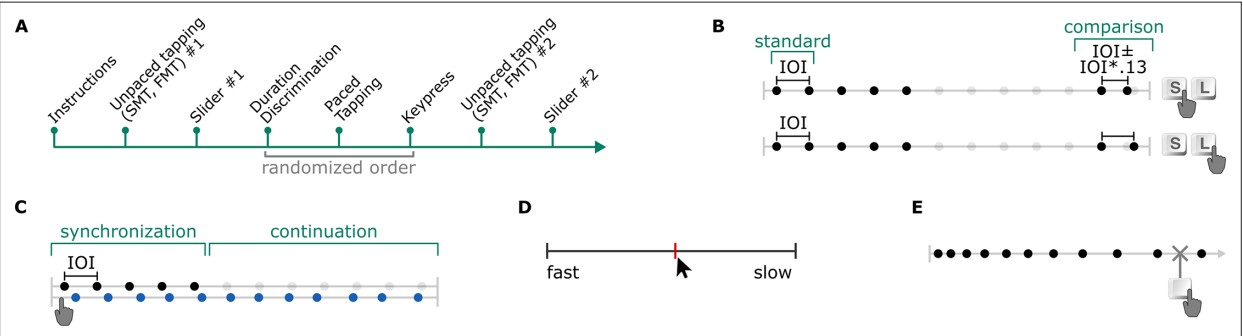

**Figure 3.** Experiment 2 (**A**) timeline, and illustrations of the (**B**) duration discrimination, (**C**) paced tapping, (**D**) slider, and (**E**) keypress tasks.

The online version of this article includes the following figure supplement(s) for figure 3:

**Figure supplement 1.** Results of the bootstrapping analysis.

Participants completed an online survey prior to the session. The lab session started with the SMT and FMT tasks, respectively. Then, participants were asked to set the sound volume to be used in the auditory tasks throughout the experiment using a slider that they clicked with a mouse. The experiment proceeded with the slider PPT task, the keypress PPT task, then the duration discrimination and paced tapping tasks, and finally with repetitions of the SMT, FMT, and slider tasks. The order of the keypress, duration discrimination and paced tapping tasks was pseudo-randomized for each participant and all six order combinations were counterbalanced. Prior to each task, participants were presented with instructions on the screen. Short breaks were allowed between tasks. Upon completion of the experiment, participants were moved to another booth in the laboratory room to complete a pure-tone audiometry measurement. An individual session including audiometry lasted 90 min on average. Instructions (see *Supplementary file 2*) were in German.

## Duration discrimination task

The stimuli for the duration discrimination task were the same as in Experiment 1. The conditions differed from Experiment 1 random-order sessions in three aspects: here, the pool of stimulus rates was linearly spaced between 200 ms and 1000 ms in 10 ms steps, comparison interval deviated from standard IOI at a fixed amount of DEV = ±13%, and there were two repetitions of each stimulus rate. For determining the spacing for IOI, we performed a bootstrapping analysis on data from our previous study, from which the current paced tapping paradigm was adapted (*Kaya and Henry, 2022*). We first downsampled each participant's single-session data from the previous study (*Kaya and Henry, 2022*) with each even step size between 4 ms and 20 ms. That is, for the respective step size, we filtered data where IOI corresponded to the spacing value added to the smallest (200 ms) to the largest (1000 ms) IOI (e.g. trials with IOI = 200, 204, 208 ms, and so on, for step size of 4 ms). We performed the preferred rate estimation procedure for each downsampled dataset, used in the experiment analyses. To assess the optimum step size that would represent the experiment's findings, we assessed the correspondences between (1) preferred rate estimates from the original and downsampled datasets for each session and (2) estimates from downsampled datasets between sessions. In both steps, the correspondence between estimates was quantified by their harmonic difference (i.e. the sum of the data points Euclidean distances to the closest line among y=x, y=2x, and y=x/2 lines). A smaller difference value indicated that the estimates subject to comparison were similar, or close to doubles or halves of each other. Harmonic differences obtained from the first and seconds steps of the bootstrapping analysis are shown in *Figure 3—figure supplement 1*. Together, the bootstrapping analyses showed that the average harmonic difference between estimates from original versus downsampled datasets was smallest at the step size of 10, where harmonic difference between downsampled sessions' estimates was also small.

We selected the fixed deviation for comparison intervals as follows. First, we estimated thresholds for negative and positive deviations from Experiment 1. To do so, for each participant's (N=27) random-order session data, we averaged the accuracy at each deviation level, separately for negative and positive deviations. We fitted psychometric curves to the mean values and obtained the deviation amount that yielded 75% predicted accuracy from the fitted curve. From the resulting distributions of thresholds for negative and positive deviations, we removed outliers by excluding any value that exceeded 3× the MAD of all threshold values in the respective distribution. Finally, we took the mean threshold value across participants and deviation directions. We then piloted the task on a small sample to confirm that the value of 13% was appropriate to be used in the duration discrimination task in Experiment 2 that would give an approximate accuracy of 75%.

The task (*Figure 3B*) consisted of two blocks with complementary DEV conditions. Participants were presented with all 81 stimulus rates in the same order in each block. However, if the comparison interval for a given stimulus rate was longer in the first block, it was shorter in the second, and vice versa. As in Experiment 1 random-order sessions, the change in IOI between consecutive trials (ΔIOI) was maximized, and the direction of the change alternated on every trial. For each participant, we generated a unique stimulus order which was constant across the blocks and was also used in the paced tapping task.

The instructions of the task included two example trials, and participants practiced the task for at least 6 trials. The properties and the procedure of the example and practice trials were identical to those in Experiment 1.

## Paced tapping task

The task (*Figure 3C*) was a shorter version of the synchronization-continuation paradigm we developed in a previous study (*Kaya and Henry, 2022*). On each trial, participants were presented with an isochronous stimulus sequence of five sounds, followed by silence. Sound stimuli were the woodblock samples used in Experiment 1. Participants were instructed to start tapping to the stimulus as soon as possible, and to continue tapping at the same rate once the sounds ceased, until the end of the trial, which was signaled by a change in the screen color. For each participant, the stimulus rates as well as their order were identical to those generated for the duration discrimination task. In these matched stimulus conditions, IOI ranged from 200 ms to 1000 ms in 10 ms steps. Allowed duration for continuation tapping was seven times the stimulus IOI for fast (IOI < 300 ms) stimuli, and six times the IOI for slow (IOI > 300 ms) stimuli. Prior to the task, participants completed 6 practice trials, with specifications described in *Kaya and Henry, 2022*.

## Unpaced tapping tasks

The procedure for the SMT task and FMT tasks was identical to those in Experiment 1.

## Slider task

The slider task was a PPT task where participants dynamically adjusted the rate of stimulus sequences comprising the same woodblock samples used in Experiment 1. Each trial started with an isochronous stimulus sequence, and participants were presented with the instructions at the top of the screen. A horizontal slider (*Figure 3D*) was displayed with labeled endpoints 'schnell' (fast) and 'langsam' (slow). Moving the mouse changed the indicator of the slider, marked in red, and each left-click produced an isochronous stimulus sequence with the selected rate. A right mouse click saved the final rate and terminated the trial. Participants completed two blocks of 8 trials of the task. In each block, the start-rate of the stimulus sequence was 200 ms in half of the trials and 1000 ms in the other half. The location of the labels also differed between trials, and the 'fast' label was on the left end in half of the trials, and vice versa in the other half. Label locations and start-rates were counterbalanced within each block, and their combinations were ordered randomly.

## Keypress task

The keypress task was also a PPT task where participants indicated their preferred rates by stopping stimulus sequences with dynamically changing rates. Stimulus samples making up the sequences were the woodblock samples used in Experiment 1. Each trial started with a stimulus sequence, and participants were presented with the instruction text on the top, and a dynamic figure on the middle of the screen that indicated the time left to respond. If no response was given during the stimulus, the trial was repeated. Stimuli started fast (IOI = 200) in half of the trials and slow (IOI = 1000) in the other half and increased or decreased by 10 ms in each interval, depending on the start-rate. That is, the stimulus got slower in each interval on fast-start trials, and vice versa on slow-start trials. Participants completed 6 trials of the keypress task. The order of the stimulus conditions was randomized. *Figure 3E* illustrates a fast-start condition of the keypress task.

## Design

The stimulus IOIs presented in all tasks that involved an auditory stimulus ranged from 200 ms to 1000 ms. Thus, IOI was an independent variable, on which rate preferences and performances were assessed to be compared across tasks. The order of stimulus IOI, and thus ΔIOI, was matched between duration discrimination and paced tapping tasks, from which independent variables of |+ΔIOI| and |−ΔIOI| were derived. Other independent variables were DEV direction (i.e. whether comparison interval was shorter or longer than the standard) in duration discrimination task, repetition for SMT, FMT, and slider tasks, and start-rate for slider and keypress tasks.

Dependent variables were the tapping rate in SMT and FMT, selected rate in slider and keypress, accuracy and bias in duration discrimination, and signed or absolute values of tempo-matching errors (TME) in paced tapping tasks.

## Apparatus

Apparatus for the presentation of sound stimuli, and collection of tapping and keyboard responses were identical to those of Experiment 1. Additionally, participants used a mouse for giving responses in the slider task, and for setting the desired sound volume. The background survey was a German translation of the survey used in Experiment 1. We conducted Experiment 2 in German given that the participant sample consisted of older individuals who were less likely to fluently speak English than the mostly student sample we recruited in Experiment 1.

## Analysis

### Data cleaning and exclusion criteria

As Experiment 2 involved multiple tasks, participants were excluded from only the respective tasks where their performance met the exclusion criteria.

The duration discrimination task in Experiment 2 had two exclusion criteria: (1) chance-level performance in both DEV directions, as in Experiment 1 and (2) ceiling performance in overall response accuracy (average accuracy >0.95). Two participants were excluded based on the first criterion, one participant was excluded based on the second.

On the trial level, the paced tapping task had two exclusion criteria: first, any inter-tap interval (ITI) that was smaller than half or bigger than 1.8 times the stimulus IOI was excluded. From the remaining ITIs, outliers were detected by the script described in *Data cleaning and exclusion criteria* for unpaced tapping tasks under *Experiment 1* in *Methods* section. On the participant level, criteria were incompatibility between stimulus rate and tapping rate, and low number of tapping intervals on average. To test the first criterion, we fitted models to overall task data where the tapping rate (i.e. the median of all ITIs in each trial after trial-level data cleaning) was predicted by stimulus IOI and obtained slopes. Two participants were excluded as they had slopes smaller than 0.5. One participant was excluded based on the second criterion, as the average number of intervals they produced across trials was smaller than 7.

The data cleaning procedure of unpaced tapping tasks was identical to that described for Experiment 1. In the slider task, we recorded whether participants listened to the different stimulus rates by clicking on the different locations on the slider. Exclusion criterion was not testing the stimulus rates on more than 75% of the trials by producing a minimum of one mouse click, which suggested that the participant did not engage with the task. One participant was excluded from the slider task based on this criterion. From the remaining participants' data, any trial without a mouse click was removed from further analyses. No exclusion criterion was defined for the keypress task.

Finally, before applying group-level statistics such as t-tests and correlations, any data point that fell outside of the interquartile range was excluded from the respective distributions.

### Outcome measures

The outcome measures from the duration discrimination task were accuracy and bias. Response coding was same as in Experiment 1. Since the duration discrimination task in Experiment 2 included two repetitions of each IOI (presented in different blocks with different DEV directions), accuracy and bias were averaged across IOI repetitions.

For each trial in the paced tapping task, we calculated the TME following the analysis in our previous study (*Kaya and Henry, 2022*). TME was the difference between tapping rate (median ITI of all taps in a trial) and stimulus IOI, normalized by stimulus IOI, described by $TME_k = ((median [ITI_1, ITI_1, \ldots, ITI_n,]) - IOI_k)/IOI_k$, where k is the trial index and n is the maximum number of intervals in a single trial. A positive TME indicated that the tapping rate was slower than stimulus rate, and a negative TME indicated that it was faster. For the unpaced tasks, the outcome measure from each trial was the tapping rate, calculated as the median ITI after trial-level data cleaning. From each trial of the SMT task, we also obtained the CV, calculated as the standard deviation of all intervals divided by their mean. We further compared SMT across repetitions of the same task throughout the experiment using Pearson correlations and paired-samples t-tests.

The slider task had two start-rate conditions and two repetitions throughout the experiment (before and after main tasks). The dependent measure for each trial was the median of all final responses. We assessed the main effects and interactions of start-rate and repetition on slider responses across participants, using a repeated-measures ANOVA. We calculated the rate preference on each trial of

the keypress task as the presented stimulus' rate at the time of the keypress. The summary measure for each start-rate was the median of all rate preferences in trials with same start-rate.

## Preferred rate estimates

Experiment 2 involved various tasks by which we aimed to estimate individuals' preferred rate. For the SMT task, we estimated preferred rate as median tapping rate. For the slider and keypress tasks (PPT), we averaged participants' indicated preference across conditions and repetitions. For both the duration discrimination and paced tapping tasks, we estimated preferred rate as the stimulus IOI yielding peak performance as follows.

Best-performance rates in the duration discrimination task were calculated by smoothing accuracy as a function of stimulus rate, as in Experiment 1. After excluding the study-specific outliers on the participant level, for each participant, we smoothed accuracy using 'Gaussian' method in *smooth-data* function in MATLAB. Following the optimization procedure used in Experiment 1, we assessed the window size that revealed a single-point maximum accuracy for each participant. The optimum window was 13 samples, which was used to smooth both the accuracy and IOI values in each participant's dataset.

The dependent measure in paced tapping task was TME, which was a signed, proportional error measure. Best-performance rates in this task were the conditions where participants tapped with the least errors, quantified by the absolute TME, |TME|. Since the paced tapping task shared the stimulus rate conditions with duration discrimination task, we used the optimum window size obtained for the duration discrimination task for smoothing |TME| so that the estimates would be maximally comparable across tasks.

## Flexibility estimates

Experiment 1 in the current study and the findings of our previous study (*Kaya and Henry, 2022*) showed robust effects of stimulus history on rhythm perception and production. As in those analyses, flexibility in Experiment 2 was defined as the ability to adapt to changes in the rhythmic context.

In the duration discrimination task, we assessed flexibility by fitting logistic models to each participant's data where accuracy was predicted either by $|-\Delta IOI|$ or $|+\Delta IOI|$, as in Experiment 1. A negative slope obtained from the models indicated that the probability of giving a correct response decreased as the $|\pm\Delta IOI|$ increased. Similarly, in the paced tapping task, we fitted linear models where |TME| was predicted either by $|-\Delta IOI|$ or $|+\Delta IOI|$. A positive slope from the models indicated that the absolute TME increased with $|\pm\Delta IOI|$. However, as a final step, we inversed the slopes obtained from this task so that more negative β estimates indicated less flexibility.

We tested the hypothesis of a decrease in oscillator flexibility with advancing age by correlating age and slopes from each $|\pm\Delta IOI|$ condition (flexibility estimates) in duration discrimination and paced tapping tasks (Pearson correlation, one-tailed). Since these analyses involved multiple comparisons, we controlled for the false discovery rate (FDR), using the Benjamini-Hochberg method (*Benjamini and Hochberg, 1995*; *Benjamini and Yekutieli, 2001*). To test whether overall performance decreased with age, we ran another series of correlations between age and average accuracy in duration discrimination task, and average |TME| in the paced tapping task, and FDR-corrected the p-values.

Additionally, we explored the relationship between individuals' age and preferred rate estimates, by separate correlation analyses between age and preferred rate estimated from each condition and measurement of the slider and keypress (PPT) tasks, and preferred rate estimates from duration discrimination and paced tapping tasks. Since we defined no hypothesis for preferred rate and age relationships, we used two-tailed Pearson correlation and no correction.

## Results

### Unpaced tapping

Tapping rates from 'fastest' and 'slowest' FMT trials showed no difference between pre- and post-session measurements and were additionally correlated across repeated measurements. Given the consistency of the measures, rates from each FMT task from first and second measurements were averaged for further analyses. Tapping rates from SMT task were also correlated across measurements. However, rates from the second measurement were significantly slower than those from the first measurement. SMT CV did not correlate across measurements (p=0.071731), and CV from the

second measurement (M=0.070, SD = 0.033) was significantly higher (t(26) = –2.5116, p=0.018563) than CV from first measurement (M=0.055, SD = 0.023). The results of the pairwise comparisons between tapping rates from all unpaced tapping tasks across measurements are provided in *Supplementary file 1b*.

## Preferred rate estimates

Individuals' PPT was measured by the slider and keypress tasks. In the slider task, rate preferences from the same start-rate conditions were significantly correlated and showed no systematic differences across repeated measurements. Within the first measurement block, rates from slow-start conditions (M=0.732, SD = 0.165) were slower than those from fast-start conditions (M=0.658, SD = 0.167) (t(25) = –2.109, p=0.045134), although they were significantly correlated (r(24) = 0.691, p=9.3667e-05). Rate preferences from the second measurement showed no difference between the start-rate conditions (p=0.70863) and were significantly correlated (r(27) = 0.521, p=0.0044391). A repeated-measures ANOVA revealed no main effects of start-rate (p=0.16985) or repetition (p=0.86523), and no interaction (p=0.06701). In the keypress task, rate preferences from the fast-start condition (M=0.467, SD = 0.092) were significantly faster than those from the slow-start condition (M=0.840, SD = 0.111) (t(28) = –13.8046, p=5.1076e-14), and we found no correlation between rate preferences across conditions (p=0.80261). The distributions of rate preferences from separate conditions of the slider and keypress tasks are shown in *Figure 4A*.

Preferred rate estimates from both the duration discrimination and paced tapping tasks, measured by the stimulus rates with best performance, correlated significantly with SMT (*Figure 4A*, see also *Supplementary file 1c*). Moreover, we found no significant differences between estimates from either task and SMT. However, estimates did not correlate between duration discrimination and paced tapping tasks (p=0.93433), and were slower (t(26) = –2.7817, p=0.0099304) in the latter (M=0.641, SD = 0.173) than in the former task (M=0.541, SD = 0.175). In *Figure 4B*, estimates from the two performance tasks and SMT (first measurement) are illustrated. In general, estimates from both the paced and unpaced tapping tasks were slower than those from the duration discrimination task. However, the nonparallel nature of the lines that connect single-participant preferred rates for each task (*Figure 4B*, left) indicates that the amount of 'slowing' in the tapping tasks relative to the discrimination task varied across individuals. We reasoned that if the degree of slowing for each individual arises from a common source for both tasks, which we will call 'the motor component', the differences between estimates for the discrimination versus both tapping tasks should be consistent. We quantified the contribution of the motor component to preferred rates from each tapping task by subtracting the duration discrimination task estimates, which yielded two difference scores (paced tapping – duration discrimination and SMT – duration discrimination). These difference scores were significantly positively correlated (r(25) = 0.54084, p=0.0035823), confirming that each individual had a consistent motor component contribution that slowed their preferred rate estimate in different tapping tasks in a similar manner.

Rate preferences in the slider task correlated with SMT only in slow-start conditions from the first measurement, and in fast-start conditions from the second measurement. Rate preferences from the keypress task only correlated with those from slider task conditions (i.e. within PPT tasks), but not with any SMT measurement or estimates from the performance tasks.

## Flexibility estimates

We hypothesized negative effects of stimulus history on performance in both perceptual and motor tasks. We found similar effects of stimulus history in both tasks. β obtained from the separate models quantifying the effect of |–ΔIOI| and |+ΔIOI| on accuracy in the duration discrimination task were both significantly smaller than zero, indicating that accuracy decreased as |±ΔIOI| increased, both in trials where the stimulus was faster and slower than previous (*Figure 5A*). In the paced tapping task, β from models where |TME| was predicted either by |+ΔIOI| or |–ΔIOI| were significantly greater than zero, indicating that TME increased as a function of |±ΔIOI| (*Figure 5B*). Paired-samples t-tests revealed no significant differences between the strength of the |–ΔIOI| vs |+ΔIOI| effect in either task. However, β from models where |+ΔIOI| predicted |TME| were numerically smaller, and significantly more variable than those models where |–ΔIOI| predicted |TME|; the difference in variability was assessed using a

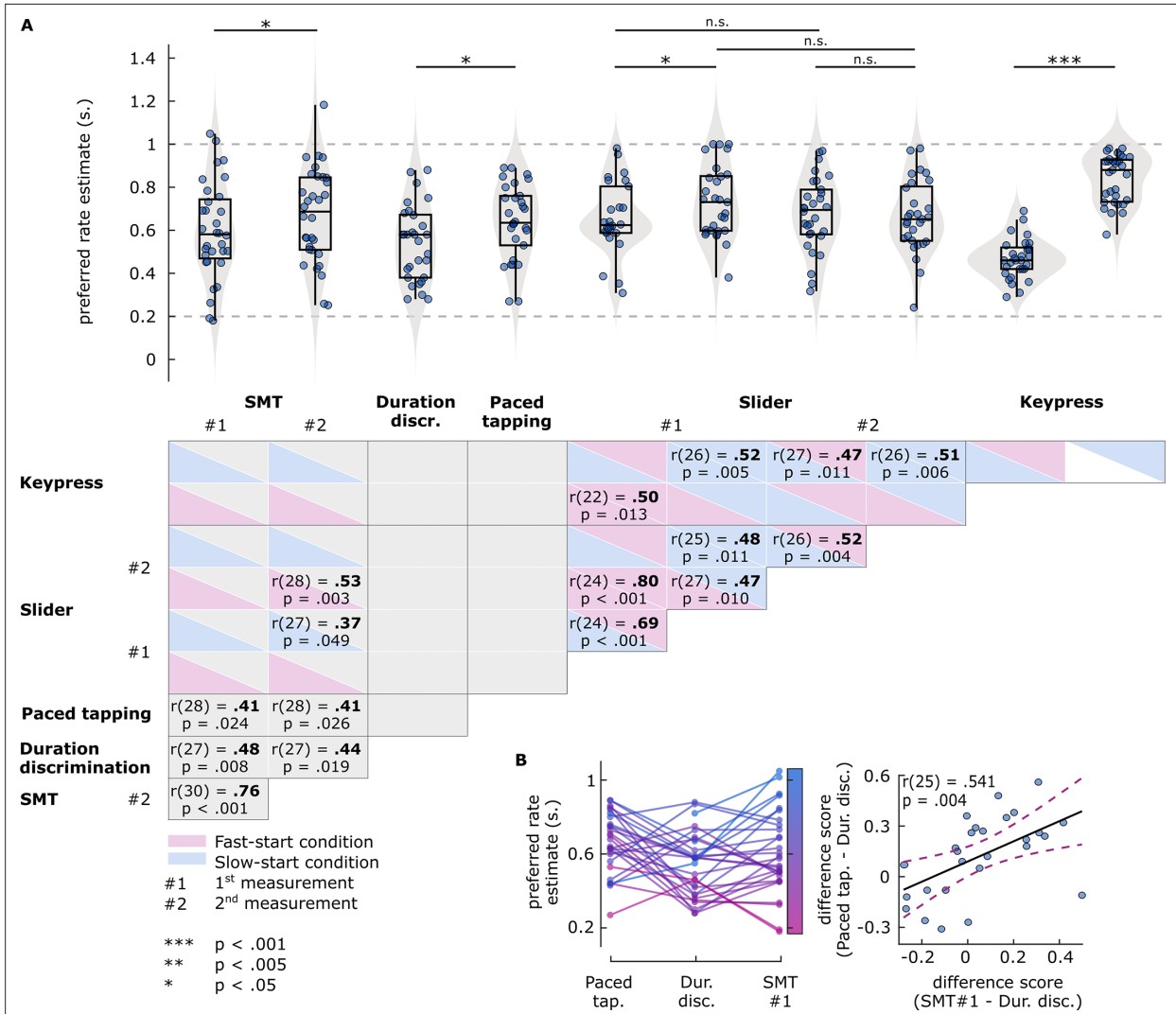

**Figure 4.** Results of Experiment 2 preferred rate analyses. (**A**) Top: Estimates of preferred rate from each task condition. Box plots show median (black vertical line), 25th and 75th percentiles (box edges), and remaining data range (whiskers). Vertical lines above the box plots represent within-participants pairwise comparisons. The horizontal dashed lines represent the minimum and maximum stimulus rates presented in the experiment. Bottom: Pairwise correlations between preferred rates across tasks. For the slider and key-press tasks, boxes are colored to indicate fast-start (pink) and slow-start (blue) conditions. Coefficients and p-values are reported for significant correlations only. (**B**) Relationship between the preferred rate estimates from the paced tapping, duration discrimination, and spontaneous motor tempo (SMT) (first measurement) tasks. Left: Participants' estimates from the three tasks. Each circle represents an individual's preferred rate estimate, connected by lines between the tasks. Both circles and lines are color-sorted by individuals' SMT, ranging from fast (pink) to slow (blue). Right: Correlation between the difference scores. Each circle represents a single participant's difference score, namely, how different the estimates from SMT (x axis) and paced tapping (y axis) tasks were than those from the duration discrimination task. Solid black line represents the regression line, dashed lines represent 95% confidence intervals.

Brown-Forsythe test ($F_{(1,54)}$ = 5.8671, p=0.01881). Descriptive statistics and test results for comparison of β estimates against zero are provided in *Supplementary file 1d*.

To investigate the direction of history effects on performance, we compared perceptual and motor biases in trials with negative and positive rate change. In conditions where the stimulus on the current trial was faster than the previous one, average bias (M=–0.166, SD = 0.094) was significantly smaller than zero ($t(28)$ = –9.4985, p=1.48e-10, *Figure 5A*); and average TME (M=0.014, SD = 0.021) was greater than zero ($t(26)$ = 3.3895, p=0.0011216, *Figure 5B*). The opposite was the case in conditions with slower-than-previous stimulus, as average bias (M=0.217, SD = 0.108) was greater ($t(27)$ = 10.587, p=2.059e-11, *Figure 5A*) and average TME (M=–0.013, SD = 0.018) was smaller ($t(26)$ = –3.7556, p=0.00044069, *Figure 5B*) than zero.

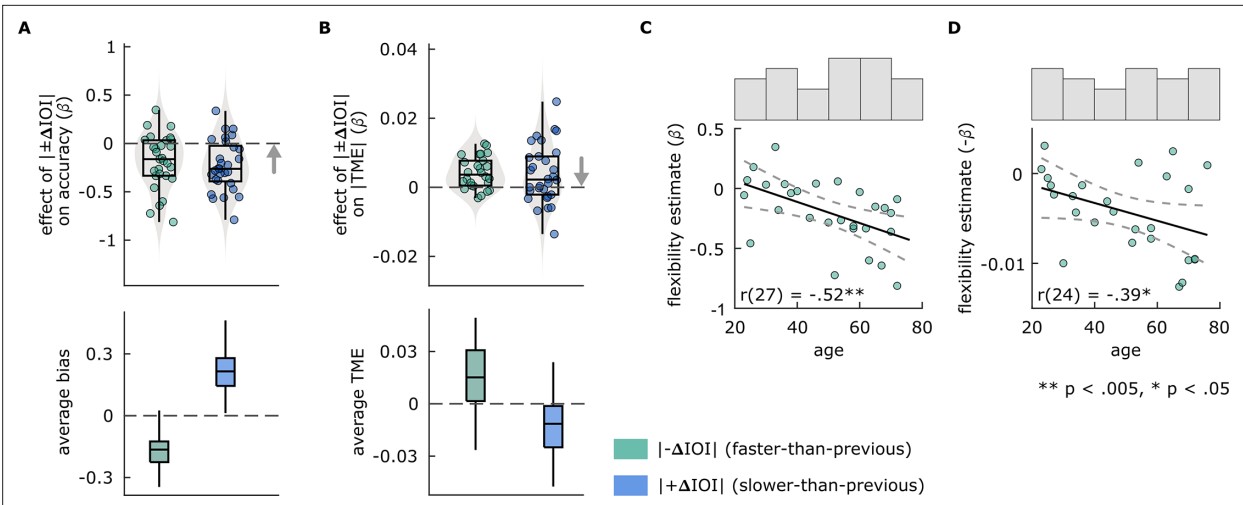

**Figure 5.** Results of Experiment 2 flexibility analyses. (**A and B**) Effects of between-trial absolute rate change (|±ΔIOI|) on performance in Experiment 2 (**A**) duration discrimination and (**B**) paced tapping tasks. In the top panels, each circle represents an individual's slope (β) obtained from models, fitted separately to conditions where |–ΔIOI| (left, green) or |+ΔIOI| (right, blue) predicted (**A**) accuracy in the duration discrimination or (**B**) |TME| in the paced tapping task. The arrow direction indicates better flexibility. In the bottom panels, box plots show (**A**) average bias in duration discrimination and (**B**) average TME in paced tapping tasks, from |–ΔIOI| (left, green) and |+ΔIOI| (right, blue) conditions. In all panels, box plots show the median (black vertical line), 25th and 75th percentiles (box edges), and extreme data points (whiskers). (**C and D**) Correlations between individuals' age and the flexibility estimates from (**C**) duration discrimination and (**D**) paced tapping tasks. Solid black lines represent the regression line, dashed lines represent 95% confidence intervals. Histograms above each plot show the distribution of participant ages after outlier corrections.

In the duration discrimination task, we also assessed the differences in responses to shorter versus longer comparison intervals as an indicator of how individuals responded to phase perturbations, by comparing accuracy in trials with |–DEV| and |+DEV|. Participants' average accuracy from the latter conditions (M=0.746, SD = 0.070) were higher (t(25) = –2.5536, p=0.017141) than those from the former conditions (M=0.694, SD = 0.116).

## Age-related changes in oscillator flexibility

One of the main goals of Experiment 2 was to compare the estimates of preferred rate and flexibility across individuals to assess the age-related changes in oscillator properties. We recruited our participant sample to have a flat age distribution, with participants ranging in age from 20 to 76 years.

The results revealed significant correlations (FDR-corrected for multiple comparisons) only between individuals' age and flexibility estimates from |–ΔIOI| conditions. β from logistic fits where |–ΔIOI| predicted accuracy in the duration discrimination task negatively correlated with age (r(27) = –0.525, p=0.0021, *Figure 5C*). Similarly, we found a significant negative correlation between the inversed β from models where |–ΔIOI| predicted |TME|, and age (r(24) = –0.389, p=0.0249, *Figure 5D*). The findings indicate that the ability to adapt to faster-than-previous rates decreased with increasing age.

## Discussion

The results of Experiment 2 revealed correspondences between preferred rate measures from various tasks, and effects of stimulus history on performance that were stronger for older individuals. The findings on preferred rate are consistent with previous research assessing tapping behavior at stimulus rates near to or far from individuals' SMT. During synchronization to (*Scheurich et al., 2018*) or continuation of (*Zamm et al., 2018*; *McAuley et al., 2006*; *Kliger Amrani and Zion Golumbic, 2020b*) a rhythmic stimulus, individuals overproduce stimulus rates that are faster, underproduce those that are slower than their SMT. During continuation tapping, produced intervals have also been shown to drift back toward individuals' SMT (*Zamm et al., 2018*; *Yu et al., 2003*). However, these previous paradigms have generally used a rough sampling of stimulus rates (e.g. 3) (*McAuley et al., 2006*; *Kliger Amrani and Zion Golumbic, 2020b*; *Yu et al., 2003*), or those that predefine conditions around SMT (*Zamm et al., 2018*; *Scheurich et al., 2018*). Here, we used a wide and finely sampled range of stimulus rates that were unrelated to individuals' SMT. Thus, that we found SMT to be the anchor rate with

optimal rhythmic performance further supports the idea that perception and production of rhythms are governed by a common mechanism which responds similarly to a range of stimulus rates across various tasks. Most work comparing individuals' timing performance across stimulus rates with respect to their SMT has made use of paradigms that involve a rhythmic motor component. The current study is the first that compared individuals' duration discrimination abilities across intervals of a rhythmic stimulus with respect to their SMT.

Preferred rates from the preference tasks with and without a rhythmic motor component (SMT and PPT, respectively) were more similar than preferred rate estimates from performance tasks (duration discrimination and paced tapping) with and without rhythmic movement. Rate preferences from the same start-rate conditions of the slider task showed strong correspondence across repeated measurements. Interestingly, rates from the fast-start conditions showed the strongest correlation across measurements, and with SMT. We interpret this difference between the fast- and slow-start conditions as being in line with the scalar property of time perception (*Wearden and Lejeune, 2008*), in that absolute timing accuracy is generally more accurate for faster rates and shorter intervals. Moreover, this finding is supported by similar findings of increased discrepancy between SMT and PPT at slow, as compared to fast stimulus rates (*Michaelis et al., 2014*). Preferred rates from the keypress task showed large differences between start-rate conditions, although rates from slow-start trials were correlated with those from most slider task conditions. Given that the keypress task involved no dynamical adjustment of stimulus rate, preferences may have been constrained to a smaller range of stimulus rates around the start-rate. Nonetheless, individual differences were still observable, and preferred rates were still consistent with those measured in the other PPT (slider) task.

Analyses focused on flexibility revealed that both duration discrimination and paced tapping performance were worse when rate change from one trial to the next was large, regardless of the direction of the change (i.e. whether stimulus was faster or slower than the previous one). In cases where stimulus in each trial was faster than the previous, slower stimulus, participants tended to perceive longer comparison intervals as shorter and tap slower than the stimulus. In the opposite cases, they tended to perceive shorter comparison intervals as longer and tap faster than the stimulus. Thus, non-zero biases and signed tapping errors observed in response to rate changes suggest that internal representations and behavior in each trial reflected the properties of the preceding trial; we will return to this point in *General discussion*. These findings are mostly in line with findings of Experiment 1 (current study) and those from our previous tapping study (*Kaya and Henry, 2022*), and further emphasize the presence of history effects on timing performance. The finding of signed tapping errors supports the idea that oscillators gradually adjust their phase and period to a newly encountered stimulus, resulting in discrepancy between the stimulus interval and oscillator period during synchronization to a rhythmic stimulus (*McAuley and Jones, 2003*; *Loehr et al., 2011*; *McAuley, 1995*). However, in our previous study (*Kaya and Henry, 2022*), tapping performance was especially affected when stimulus rates were faster than the preceding trial. In that study, |TME| was calculated from only synchronization tapping for the flexibility analysis. Here, we calculated |TME| from all taps from both the synchronization and continuation segments of each trial due to the lower number of trials. That is, in our previous study, we focused only on the first produced intervals on each trial, whereas here we included intervals that were produced after participants had a longer period to adapt to the new stimulus rate.

A critical finding from the current study was that flexibility, estimated inversely from the strength of the effect of |–ΔIOI| on performance in both tasks with and without a motor component, decreased with age. Reduced performance in timing tasks for aging individuals is a common finding across perceptual (*Szymaszek et al., 2009*; *Incao et al., 2022*; *Henry et al., 2017*) and motor (*Turgeon et al., 2011*; *von Schnehen et al., 2022*) tasks. However, overall timing performance measures, namely, task averages of duration discrimination accuracy and tapping errors, showed no systematic relationships with individuals' age, suggesting that age-related changes in rhythm perception might be specific to adaptive mechanisms rather than general timing abilities.

In addition to focusing on deviations in stimulus rate between trials, we also assessed how participants responded to within-trial deviations, i.e., how much comparison interval deviated from the stimulus IOI. As in Experiment 1, however, significantly here, accuracy was marginally higher in conditions with longer compared to shorter comparison intervals. That this difference reached significance only in the current study may be due to the age of the participant sample, given the finding that adapting to faster, but not slower stimulus was more challenging for older individuals.

Of note is that the paradigm in Experiment 2 was derived from two multi-session experiments through a series of reliability and bootstrapping analyses. The longer versions of the duration discrimination (Experiment 1, current study) and paced tapping (synchronization-continuation paradigm in *Kaya and Henry, 2022*) involved around 400 trials in each of the two sessions, between which the estimates of preferred rate and flexibility were also consistent. Thus, the current paradigm can be used to assess internal oscillator properties in clinical settings or with participant samples where concerns for task difficulty or fatigue may arise.

## General discussion

The goal of the current set of studies was to highlight factors that impact auditory rhythm processing. To this end, we conducted two experiments, investigating the interplay between the properties of the external world (the stimulus) and the individual responding to the stimulus (the perceiver). Adopting an entrainment perspective that considers internal oscillators as the underlying mechanism for rhythm processing (*Large and Jones, 1999*; *McAuley, 2010*), we aimed to capture this interplay by characterizing the properties of internal oscillators, and to assess how they change with advancing age. Specifically, we estimated oscillators' preferred rates and flexibility for each individual in perceptual and motor tasks, assessed the relationship between rate preferences and optimal stimulus rates for timing performance, and tested the hypothesis that oscillator flexibility diminishes as we age.

Experiment 1 was a perceptual paradigm, where individuals' ability to discriminate between stimulus intervals over a wide range of finely sampled stimulus rates was assessed in two temporal contexts: one that required rapid temporal adaptation, challenging oscillator flexibility, and one without such requirement. In Experiment 2, we combined shorter versions of the duration discrimination paradigm (Experiment 1) and a paced tapping paradigm (adapted from *Kaya and Henry, 2022*), using matching stimulus conditions. Experiment 2 also involved a common measure of preferred rate, the SMT task, and two 'PPT tasks (slider, keypress) where individuals' rate preferences were assessed. From the performance paradigms, we estimated preferred rate as the stimulus rates with best performance, indexed by maximum accuracy in the duration discrimination tasks, and minimum tempo-matching errors (TME) in the paced tapping task. We defined flexibility as the ability to adapt to changes in stimulus rate, which was inversely related to how much single-trial performance was affected by trial-to-trial changes in stimulus rate.

### Preferred rate estimates

In the rhythmic entrainment literature, preferred rate is typically estimated by SMT. However, two main aspects of the SMT task motivated us to question its explanatory power for predicting individuals' perceptual abilities in real-world listening situations. First, given that the task involves periodic motor actions, the relative contributions of an internal timekeeper versus constraints or resonances of an individual's motor system to the produced tapping rate cannot be separated. Second, SMT is a preference measure, since it measures the rate at which individuals prefer to tap at, without introducing any interaction with a stimulus. Although there is evidence for positive relationships between SMT and rates yielding best timing abilities in paced tapping tasks (*Zamm et al., 2018*; *Scheurich et al., 2018*), rate preferences obtained from SMT task may not necessarily predict how individuals would perform at other auditory tasks, especially those that don't involve periodic motor actions. Here, we aimed to bridge this gap and understand the potential predictive power of SMT for perceptual performance situations with higher ecological validity, by directly comparing SMT to 'performance' measures of preferred rate both with and without a motor component. Based on the assumptions of entrainment models, we estimated preferred rate as the stimulus rate with peak performance. Findings from Experiment 1 validated this estimation method by showing that accuracy in single-session datasets not only peaked at this stimulus rate, but also systematically increased toward this value, consistent with predictions based on detuning.

The results of Experiment 2 revealed that the stimulus rates for which individuals showed better timing performance were indeed correlated with SMT. However, we did not find one-to-one correspondences between SMT and preferred rate estimates from the performance tasks, and estimates were not correlated across the performance tasks. SMT was more variable across participants than preferred rates estimated from either of the performance tasks, and preferred rates estimated from

tasks involving a motor component (SMT, paced tapping) tended to be slower than those estimated from the duration discrimination task. We discuss two possible primary dimensions along which these tasks differ and how these might preclude directly predicting performance on one task based on the rate preference for another: involvement of the motor system and indicating preference versus interacting with an environmental rhythm.

Both the unpaced (SMT) and paced tapping tasks required rhythmic motor responses, as compared to the duration discrimination task where perceptual judgments were assessed. We found that preferred rate estimates from both motor tasks were slower than those obtained via duration discrimination. Interestingly, we found that the degree of 'slowing down' in the motor compared to the discrimination tasks was consistent within an individual. This suggests that the contribution of the 'motor component' to preferred rate is individually specific and quantifiable. This finding is in line with the proposal that perception and production of rhythms are governed by a system of multiple coupled oscillators (*Zalta et al., 2020*; *Assaneo et al., 2021*), with the observed preferred rate in any task being jointly influenced by preferred rate of a perceptual (in this case, auditory) oscillator, preferred rate of a motor oscillator, and the coupling strength between these two nodes. Indeed, similar discrepancies between preferred rates of auditory and motor oscillators were observed in speech comprehension and were attributed to individual differences in auditory-motor coupling (*Lubinus et al., 2023*). Under this assumption, we propose that the differences between preferred rate estimates from tasks with and without tapping (motor) responses, i.e., the degree of slowing when the motor component is added, will increase with the difference in eigenfrequencies of the perceptual and motor oscillators (their detuning), and decrease with increasing coupling strength.

The other difference between the tasks by which preferred rate was estimated was the requirement to interact with a stimulus rhythm in the performance tasks, whereas the SMT and PPT tasks only involved indicating a preference. *Jones and Mcauley, 2005* argue that in the presence of a stimulus, the preferred rate can be 'pushed around' by the temporal context, given that the oscillators are adaptive and can perform within their entrainment regions. Results of Experiment 1 confirmed this prediction by revealing an effect of temporal context on preferred rate: the distribution of estimates from the temporally challenging condition was narrower than that from the condition that required minimal temporal adaptation. Thus, stimulus presentation in Experiment 2 duration discrimination and paced tapping tasks as opposed to SMT task may have contributed to the differences in preferred rate estimates. Additionally, in the paced tapping task, participants synchronized to the stimulus, which is shown to improve performance in tapping precision (*Kliger Amrani and Zion Golumbic, 2022*; *Schmidt-Kassow et al., 2013*) and perceptual judgments (*Manning and Schutz, 2013*; *Manning et al., 2017*), and thus may have contributed to the estimate differences.

## Flexibility estimates

One main goal of the current study was to investigate the circumstances that negatively impact timing abilities. Specifically, we focused on trial-to-trial changes in stimulus rate, and to what extent individuals were able to adapt to such changes, which was our definition of oscillator flexibility. In line with previous literature which reveals effects of stimulus history on perceptual (*Wiener et al., 2014*; *Jones and Mcauley, 2005*; *Wiener and Thompson, 2015*; *McAuley and Miller, 2007*) and motor (*Scheurich et al., 2020*; *Kaya and Henry, 2022*; *Motala et al., 2020*; *Large et al., 2002*; *Loehr et al., 2011*) responses, results of the current study showed that performance in duration discrimination and paced tapping tasks decreased as trial-to-trial changes in stimulus rate increased. Moreover, single-trial responses were biased such that they reflected the properties of the stimulus from the preceding trial. This set of findings is in line with predictions of oscillator models (*McAuley, 1995*). In a changing rhythmic context, the oscillator adapts to the newly encountered stimulus rate by gradually updating its phase and period (*McAuley and Jones, 2003*). The extent and time course of adaptation, however, will depend on the oscillator's flexibility, which might be modeled via error correction parameters in commonly used models of interval timing (*McAuley and Jones, 2003*; *McAuley, 1995*) or synchronized tapping (*Loehr et al., 2011*). An inflexible oscillator's period would adjust more slowly to a new rate, and so would continue to reflect the previously entrained rate, due to hysteresis. For the duration discrimination task, any comparison interval that is shorter than the oscillator's period would be classified as 'shorter', and vice versa, regardless of whether the interval was indeed shorter than the intervals making up the standard, isochronous rhythm. This means that when the previous trial was faster

than the current one, the oscillator period would be relatively short, and participants would be biased to judge comparisons as 'longer'. Conversely, when the previous trial was slower than the current one, the oscillator period would be relatively long, and 'shorter' responses would be more likely. The analysis of bias indicated that this was exactly the case for the current data. Similarly, tapping rates gradually updated from the preceding stimulus rate to a current one, resulting in TME in the direction of the previous stimulus rate. That is, when the previous trial was faster than the current one, tapping rates would underestimate the stimulus rate, and when the previous trial was slower than the previous one, tapping rates would overestimate the stimulus rate. Again, the TME analysis confirmed this to be the case. Another theoretical approach to oscillator flexibility concerns how the oscillator responds to situations with varying amounts of detuning (i.e. the difference between stimulus rate and preferred rate). Flexible oscillators can synchronize to wider ranges of stimulus rates around their preferred rate than inflexible ones. In other words, detuning does not constitute a strong determinant of a *flexible* oscillator's synchronization abilities. Results from Experiment 1 that showed negative correlations between flexibility estimates and relative-detuning slopes reveal compatibility with this detuning-based approach to oscillator flexibility. These findings suggest that the increase in accuracy toward an oscillator's preferred rate depended more strongly on detuning for inflexible oscillators, whereas synchronization abilities of the flexible ones were less dependent on detuning.

## Age-related changes in oscillator flexibility

A critical finding of the current study was an age-related decline in a specific ability: temporal adaptation to faster-than-previous stimuli. In trials where the stimulus was faster than the previous one, accuracy in the duration discrimination task decreased, and TME in the paced tapping task increased as a function of the amount of rate difference between trials, more so for older individuals.

The timing literature reveals age-related changes in time perception, such as a decrease in the accuracy of temporal estimates (*Xu and Church, 2017*), and slower tapping rates in spontaneous (*McAuley et al., 2006*; *Baudouin et al., 2004*; *Vanneste et al., 2001*) or forced (*Turgeon et al., 2011*) unpaced tapping tasks. These changes are generally attributed to slowing of the internal timekeeper mechanisms (*Baudouin et al., 2004*; *Szymaszek et al., 2009*) or a reduction of attentional resources (*Lustig and Meck, 2001*). Moreover, studies comparing older and younger individuals' preferences and performances in paced tapping paradigms reveal mixed results (*von Schnehen et al., 2022*). In the current study, we did not observe age-related changes in overall performance measures such as perceptual accuracy or tapping errors, and contrary to previous work we did not find a slowing of preferred rate no matter how it was estimated. Instead, these findings rather point to age-related changes in adaptive mechanisms underlying temporal processing. Studies assessing temporal adaptation abilities show that older individuals adapt their movements to temporal perturbations more slowly and less efficiently than younger individuals (*King et al., 2013*; *Wolpe et al., 2020*) and with less error correction (*Pollok et al., 2022*). We observed an age-related decline in temporal adaptation during both perception of and synchronization with auditory stimuli, suggesting a common source that affected the two means of responding.

Previous work reveals age-related differences in neural entrainment to auditory rhythms. Most studies focused on neural entrainment to amplitude modulated sounds show that older adults entrain *more* strongly and in a more stereotyped (less flexible) way to metronomic stimuli like those we used here (*Goossens et al., 2016*; *Herrmann et al., 2019*; *Purcell et al., 2004*). A similar pattern was observed for entrainment to the amplitude envelope of speech (*Decruy et al., 2020*; *Presacco et al., 2016*). A mixed pattern of results were reported for frequency modulated sounds; however, the existing data suggest that these differences might depend on parameters such as modulation rate and depth (*Henry et al., 2017*; *Boettcher et al., 2002*), which we will not further address here. Moreover, older adults show less neural adaptation than younger adults in temporal contexts where stimulus rate changes gradually and predictably (*Herrmann et al., 2019*). Another functional difference between younger and older brains, potentially relevant here, are findings on 'neural noise'. Variability in brain activity as measured in the BOLD signal using functional magnetic resonance imaging is higher in younger than older brains, again suggesting inflexible and stereotyped neural activity. Indeed, neural noise is associated with faster and more consistent performance across a variety of cognitive tasks (*Garrett et al., 2011*; *Grady and Garrett, 2014*). Similarly, 1/f noise measured with EEG, associated with predictive processing in a lexical task, was lower for older than younger individuals (*Dave et al.,*

*2018*). Taken together, these results suggest that poorer performance in temporal tasks that involve prediction and adaptation might reflect less flexible, overly stereotyped neural responses in older adults. This might indicate a loss of flexibility in the generating oscillator(s).

An interesting aspect of the current findings was that adaptation to faster, but not slower stimulus rates, was more difficult for older individuals. Oscillator models predict this asymmetry, with increased tapping asynchronies to speeding up compared to slowing down stimuli due to the 'period adaptation function' of the oscillator (*Loehr et al., 2011*). This was the case for the paced tapping paradigm (current study), as the effect of rate change on tapping errors was smaller and significantly more variable when stimuli slowed down as opposed to sped up, paralleling our previous findings (*Kaya and Henry, 2022*). In the duration discrimination tasks, although the magnitude of the effect of rate change was similar for both rate-change directions, only adaptation to faster stimuli worsened with age. Though evidence shows reduced adaptation to time-compressed (*Peelle and Wingfield, 2005*) or artificially speeded (*Schneider et al., 2005*) speech in older individuals, further research is needed to address the sources of adaptation to fast versus slow stimuli in aging.

## Individual differences in internal oscillator properties

One advantage of the current approach is its focus on individual variability. Previous work on rhythm perception and production, as well as aging, has largely used traditional statistical approaches involving group or condition comparisons of central tendency measures. In these cases, variability is attributed to measurement error or noise. In the current work, we opted to view variability as potentially attributable to individual differences in internal oscillator properties that may in future work be shown to have predictive power for successful outcomes in real-world listening situations. Taking this approach focused on individual differences revealed several novel findings that would have otherwise not been accessible. First, we found correspondence between the rates individuals prefer to tap their finger at, listen to, and perform perceptual and motor tasks most accurately, all pointing to preferred rates of potentially coupled, perceptual, and motor internal oscillatory systems. Second, we observed harmonic relationships between the preferred rates estimated from the duration discrimination paradigm under two different temporal contexts (Experiment 1). this is in line with the assumption that oscillators are capable of entraining to multiple stimulus rates within a temporal hierarchy (*McAuley, 1995*; *Large, 2008*), and further strengthens our choice to adopt an entrainment approach here. Finally, we found that oscillator flexibility decreased with age; this finding is supported by evidence from neural entrainment research and adds to the narrative regarding the effects of aging on the auditory system.

The pared-down versions of the duration discrimination and paced tapping paradigms described in Experiment 2 were carefully designed based on the analyses of their correspondence between Experiment 1 and our previous tapping study (*Kaya and Henry, 2022*) in terms of their main results. That is, we designed the Experiment 2 tasks to be the streamlined versions that would yield the same main results as their longer counterparts. The reasons for minimizing the duration of the tasks were (1) it allowed us to test and compare perception and production in a within-participant manner in a single session, and (2) it improved suitability for testing older adults, who we did not want to subject to an overly long or multi-session experiment. That the results of Experiment 2 replicated those from Experiment 1 and *Kaya and Henry, 2022*, independently confirmed the robustness of the designs. Thus, we would propose that these minimized designs could be used in a more diagnostic capacity in future work to measure and test predictions about internal oscillator properties of older adults or a clinical population of interest.

## Conclusion

To summarize, we adopted an entrainment approach to rhythm perception and production, which proposes that these abilities are governed by internal oscillatory mechanisms. We then developed a paradigm to estimate individuals' internal oscillator properties based on the common assumptions of the entrainment models. Performance in both duration discrimination and synchronized tapping tasks was best at a range of stimulus rates that was specific to each individual – their preferred rate – and was broadly consistent with preferred rates estimated from preference tasks (SMT). One important departure from this consistency was that involving a motor requirement slowed preferred rates, and we were able to quantify the contribution of this motor component, which was consistent within

individuals across different tasks. Performance decreased as a function of change in stimulus rate between consecutive trials. The extent to which individuals were able to adapt to the changes – oscillator flexibility – decreased with age, in accordance with research on neural entrainment and neural noise.

Several aspects of the current findings speak against alternative explanations of timekeeper models. First, an increase in performance at certain stimulus rates that show consistency across multiple measurements (Experiment 1) and tasks with and without a motor component (Experiment 2) is predicted by entrainment models (Assumption 3), but not timekeeper theories as the latter models assume a flat performance profile across stimulus rates, following 'Weber's law' (*Gibbon et al., 1984*; *Gibbon, 1977*). Second, we observed systematic increases in task accuracy (Experiment 1) toward the best-performance rates (i.e. preferred rate estimates), with the steepness of this increase being closely related to the effects of rate change (i.e. oscillator flexibility). Two interdependent properties of an underlying system together modulating an individual's timing responses show strong support for the entrainment approach. Moreover, preferred rate estimates showed harmonic relationships across multiple measurements, which is compatible with the properties of oscillator models (Assumption 4), and not predicted by timekeeper models. Finally, studies adopting a timekeeper approach suggest that timing responses should gravitate toward the mean of the presented stimulus rates in a given experimental session (*Jazayeri and Shadlen, 2010*), which should have resulted in similar patterns of results in the two sessions of Experiment 1, where only the trial order differed. We found significant accuracy and bias differences between the sessions that cannot be solely attributed to the gravitation toward the mean as the temporal statistics for the stimuli were identical across sessions.

Overall, these findings support the general hypothesis of DAT that an oscillatory system with a stable preferred rate underlies rhythm perception and production. We further show that this system loses its ability to flexibly adapt to changes in the external rhythmic context as we age.

## Acknowledgements

This work is supported by a European Research Council (ERC) Starting Grant (BRAINSYNC-804029) and a Max Planck Research Group awarded to MJH. The authors thank Kristin Weineck for helping with German translations and Paola Najera Maldonado for her support in data collection.

## Additional information

### Funding

| Funder | Grant reference number | Author |
| --- | --- | --- |
| European Research Council | BRAINSYNC-804029 | Molly J Henry |
| Max-Planck-Gesellschaft | Max Planck Research Group | Molly J Henry |
| Institute for Advanced Studies at Aix-Marseille University Fellowship 2023-24 | | Sonja A Kotz |

The funders had no role in study design, data collection and interpretation, or the decision to submit the work for publication. Open access funding provided by Max Planck Society.

### Author contributions

Ece Kaya, Conceptualization, Formal analysis, Validation, Investigation, Visualization, Methodology, Writing - original draft, Project administration, Writing – review and editing; Sonja A Kotz, Conceptualization, Supervision, Methodology, Writing – review and editing, Discussion; Molly J Henry, Conceptualization, Resources, Formal analysis, Supervision, Funding acquisition, Validation, Investigation, Methodology, Project administration, Writing – review and editing

## Author ORCIDs

Ece Kaya ⓘ http://orcid.org/0000-0003-4012-9469
Sonja A Kotz ⓘ https://orcid.org/0000-0002-5894-4624
Molly J Henry ⓘ http://orcid.org/0000-0002-2284-8884

## Ethics

"Written informed consent was obtained from all persons who participated in Experiment 1 and Experiment 2. The procedures of the experiments were approved by the Ethics Council of the Max Planck Society (approval number 2019_04) and the Research Ethics Board at Toronto Metropolitan University in accordance with the Declaration of Helsinki."

Reviewer #1 (Public review): https://doi.org/10.7554/eLife.90735.4.sa1
Reviewer #2 (Public review): https://doi.org/10.7554/eLife.90735.4.sa2
Author response https://doi.org/10.7554/eLife.90735.4.sa3

---

# Additional files

## Supplementary files

• Supplementary file 1. Supplementary tables. (a) Descriptive statistics and test results for comparison of β estimates against null distributions in Experiment 1 analyses. (b) Descriptive statistics of unpaced tapping measures in first and second experiments, and test results for pairwise comparisons. (c) Results of the pairwise correlation analyses between preferred rate estimates from each task and condition in Experiment 2. (d) Descriptive statistics and test results for comparison of β estimates against null distributions in Experiment 2 analyses.

• Supplementary file 2. Experiment 2 instructions.

• MDAR checklist

## Data availability

Experiment software, raw data obtained from Experiments 1 and 2 and analysis codes has been uploaded to OSF (https://osf.io/2vfsp).

The following dataset was generated:

| Author(s) | Year | Dataset title | Dataset URL | Database and Identifier |
|-----------|------|---------------|-------------|-------------------------|
| Kaya E | 2023 | Supplemental materials for preprint: A novel method for estimating properties of attentional oscillators reveals an age-related decline in flexibility | https://doi.org/10.17605/OSF.IO/2VFSP | Open Science Framework, 10.17605/OSF.IO/2VFSP |

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
