## [Editor Report · eLife assessment]

This **important** study has practical and theoretical implications for understanding rhythm perception and production in human cognition. The evidence for individual frequency preferences and a deterioration in frequency adaptation with age is **convincing**. These findings will inform existing models of rhythm perception and production, and the reported effects of age may have clinical implications.

---

## [Referee Report · Reviewer #1 (Public review)]

Summary:

This study assumes but also demonstrates that auditory rhythm processing is produced by internal oscillating systems and evaluates the properties of internal oscillators across individuals. The authors designed an experiment and performed analyses that address individuals' preferred rate and flexibility, with a special focus on how much past rhythms influence subsequent trials. They find evidence for such historical dependence and show that we adapt less well to new rhythms as we age. Furthermore, the revised version of this manuscript includes evidence for detuning; i.e., a gradual reduction in accuracy as the difference between a participant's preferred rate and stimulus rate increases. Such detuning also correlates with modelled oscillator flexibility measures. Such outcomes increase our credence that an entrainment-based interpretation is indeed warranted. Regardless of mechanism though, this work contributes to our understanding of individual differences in rhythm processing.

Strengths:

The inclusion of two tasks -- a tapping and a listening task -- complement each other methodologically. By analysing both the production and tracking of rhythms, the authors emphasize the importance of the characteristics of the receiver, the external world, and their interplay. The relationship between the two tasks and components within tasks are explored using a range of analyses. The visual presentation of the results is very clear. The age-related changes in flexibility are useful and compelling. The paper includes a discussion of the study assumptions, and it contextualizes itself more explicitly as taking entrainment frameworks as a starting point. Finally, the revised versions show creative additional analyses that increase our credence in an entrainment-based interpretation versus an interpretation of timekeeper other models, increasing the theoretical relevance of this study as compared to previous work.

Weaknesses:

The authors have addressed many of the weaknesses of previous peer review rounds. One final point is that our credence in an entrainment-based interpretation of these results could further increase by not only carefully outlining what is expected under entrainment (as is now done), but to also specify more extensively what predictions emerge from a timekeeper or other model, and how these data do not bear out such predictions.

---

## [Referee Report · Reviewer #2 (Public review)]

Summary:

The current work describes a set of behavioral tasks to explore individual differences in the preferred perceptual and motor rhythms. Results show a consistent individual preference for a given perceptual and motor frequency across tasks and, while these were correlated, the latter is slower than the former one. Additionally, the adaptation accuracy to rate changes is proportional to the amount of rate variation and, crucially, the amount of adaptation decreases with age.

Strengths:

Experiments are carefully designed to measure individual preferred motor and perceptual tempo. Furthermore, the experimental design is validated by testing the consistency across tasks and test-retest, what makes the introduced paradigm a useful tool for future research.

The obtained data is rigorously analyzed using a diverse set of tools, each adapted to the specificities across the different research questions and tasks.

This study identifies several relevant behavioral features: (i) each individual shows a preferred and reliable motor and perceptual tempo and, while both are related, the motor is consistently slower than the pure perceptual one; (ii) the presence of hysteresis in the adaptation to rate variations; and (iii) the decrement of this adaptation with age. All these observations are valuable for the auditory-motor integration field of research, and they could potentially inform existing biophysical models to increase their descriptive power.

Weaknesses:

To get a better understanding of the mechanisms underlying the behavioral observations, it would have been useful to compare the observed pattern of results with simulations done with existing biophysical models. However, this point is addressed if the current study is read along with this other publication of the same research group: Kaya, E., & Henry, M. J. (2024, February 5). Modeling rhythm perception and temporal adaptation: top-down influences on a gradually decaying oscillator. https://doi.org/10.31234/osf.io/q9uvr

---

## [Author Response]

The following is the authors’ response to the previous reviews.

We thank the reviewers for their thorough re-evaluation of our revised manuscript. Addressing final issues they raised has improved the manuscript further. We sincerely appreciate the detailed explanations that the reviewers provided in the "recommendations for authors" section. This comprehensive feedback helped us identify the sources of ambiguity within the analysis descriptions and in the discussion where we interpreted the results. Below, you will find our responses to the specific comments and recommendations.

**Reviewer #1 (Recommendations):**
(1) I find that the manuscript has improved significantly from the last version, especially in terms of making explicit the assumptions of this work and competing models. I think the response letter makes a good case that the existence of other research makes it more likely that oscillators are at play in the study at hand (though the authors might consider incorporating this argumentation a bit more into the paper too). Furthermore, the authors' response that the harmonic analysis is valid even when including x=y because standard correlation analysis were not significant is a helpful response. The key issue that remains for me is that I have confusions about the additional analyses prompted by my review to a point where I find it hard to evaluate how and whether they demonstrate entrainment or not.First, I don't fully understand Figure 2B and how it confirms the Arnold tongue slice prediction. In the response letter the authors write: "...indicating that accuracy increased towards the preferred rate at fast rates and decreased as the stimulus rate diverged from the preferred rate at slow rates". The figure shows that, but also more. The green line (IOI < preferred rate) indeed increases toward the preferred rate (which is IOI = 0 on the x-axis; as I get it), but then it continues to go up in accuracy even after the preferred rate. And for the blue line, performance also continues to go up beyond preferred rate. Wouldn't the Arnold tongue and thus entrainment prediction be that accuracy goes down again after the preferred rate has passed? That is to say, shouldn't the pattern look like this (https://i.imgur.com/GPlt38F.png) which with linear regression should turn to a line with a slope of 0?This was my confusion at first, but then I thought longer about how e.g. the blue line is predicted only using trials with IOI larger than the preferred rate. If that is so, then shouldn't the plot look like this? (https://i.imgur.com/SmU6X73.png). But if those are the only data and the rest of the regression line is extrapolation, why does the regression error vary in the extrapolated region? It would be helpful if the authors could clarify this plot a bit better. Ideally, they might want to include the average datapoints so it becomes easier to understand what is being fitted. As a side note, colours blue/green have a different meaning in 2B than 2D and E, which might be confusing.

We thank the reviewer for their recommendation to clarify the additional analyses we ran in the previous revision to assess whether accuracy systematically increased toward the preferred rate estimate. We realized that the description of the regression analysis led to misunderstandings. In particular, we think that the reviewer interpreted (1) our analysis as linear regression (based on the request to plot raw data rather than fits), whereas, in fact, we used *logistic* regression, and (2) the regression lines in Figure 2B as *raw* IOI values, while, in fact, they were the *z-scored* IOI values (from trials where stimulus IOI were faster than an individual’s preferred rate, IOI < preferred rate, in green; and from trials stimulus IOI were slower than an individual’s preferred rate, IOI > preferred rate, in blue), as the x axis label depicted. We are happy to have the opportunity to clarify these points in the manuscript. We have also revised Figure 2B, which was admittedly maybe a bit opaque, to more clearly show the “Arnold tongue slice”.

The logic for using (1) logistic regression with (2) Z-scored IOI values as the predictor is as follows. Since the response variable in this analysis, accuracy, was binary (correct response = 1, incorrect response = 0), we used a logistic regression. The goal was to quantify an acrosssubjects effect (increase in accuracy toward preferred rate), so we aggregated datasets across all participants into the model. The crucial point here is that each participant had a different preferred rate estimate. Let’s say participant A had the estimate at IOI = 400 ms, and participant B had an estimate at IOI = 600 ms. The trials where IOI was faster than participant A’s estimate would then be those ranging from 200 ms to 398 ms, and those that were slower would range from 402 ms to 998 ms. For Participant B, the situation would be different: trials where IOI was faster than their estimate would range from 200 ms to 598 ms, and slower trials would range between 602 ms to 998 ms. For a fair analysis that assesses the accuracy increase, regardless of a participant’s actual preferred rate, we normalized these IOI values (faster or slower than the preferred rate). Zscore normalization is a common method of normalizing predictors in regression models, and was especially important here since we were aggregating predictors across participants, and the predictors ranges varied across participants. Z-scoring ensured that the scale of the sample (that differs between participant A and B, in this example) was comparable across the datasets. This is also important for the interpretation of Figure 2B. Since Z-scoring involves mean subtraction, the zero point on the Z-scaled IOI axis corresponds to the mean of the sample prior to normalization (for Participant A: 299 ms, for Participant B: 399 ms) and *not* the preferred rate estimate. We have now revised Figure 2B in a way that we think makes this much clearer.

The manuscript text includes clarification that the analyses included logistic regression and stimulus IOI was z-scored:

“In addition to estimating the preferred rate as stimulus rates with peak performance, we investigated whether accuracy increased as a function of detuning, namely, the difference between stimulus rate and preferred rate, as predicted by the entrainment models (Large, 1994; McAuley, 1995; Jones, 2018). We tested this prediction by assessing the slopes of mixed-effects logistic regression models, where accuracy was regressed on the IOI condition, separately for stimulus rates that were faster or slower than an individual’s preferred rate estimate. To do so, we first z-scored IOIs that were faster and slower than the participant’s preferred rate estimates, separately to render IOI scales comparable across participants.” (p. 7)

While thinking through the reviewer’s comment, we realized we could improve this analysis by fitting mixed effects models separately to sessions’ data. In these models, fixed effects were z-scored IOI and ‘detuning direction’ (i.e., whether IOI was faster or slower than the participant’s preferred rate estimate). To control for variability across participants in the predicted interaction between z-scored IOI and direction, this interaction was added as a random effect.

“Ideally, they might want to include the average datapoints so it becomes easier to understand what is being fitted.”

Although we agree with the reviewer that including average datapoints in a figure in addition to model predictions usually better illustrates what is being fitted than the fits alone, this doesn’t work super well for logistic regression, since the dependent variable is binary. To try to do a better job illustrating single-participant data though, we instead fitted logistic models to each participant’s single session datasets, separately to conditions where z-scored IOI from fasterthan-preferred rate trials, and those from slower-than-preferred rate trials, predicted accuracy. From these single-participant models, we obtained slope values, we referred to as ‘relative detuning slope’, for each condition and session type. This analysis allowed us to illustrate the effect of relative detuning on accuracy for each participant. Figure 2B now shows each participant’s best-fit lines from each detuning direction condition and session.

Since we now had relative detuning slopes for each individual (which we did not before), we took advantage of this to assess the relationship between oscillator flexibility and the oscillator’s behavior in different detuning situations (how strongly leaving the preferred rate hurt accuracy, as a proxy for the width of the Arnold tongue slice). Theoretically, flexible oscillators should be able to synchronize to wide range of rates, not suffering in conditions where detuning is large (Pikovsky et al., 2003). Conversely, synchronization of inflexible oscillators should depend strongly on detuning. To test whether our flexibility measure predicted this dependence on detuning, which is a different angle on oscillator flexibility, we first averaged each participant’s detuning slopes across detuning directions (after sign-flipping one of them). Then, we assessed the correlation between the average detuning slopes and flexibility estimates, separately from conditions where |-𝚫IOI| or |+𝚫IOI| predicted accuracy. The results revealed significant negative correlations (Fig. 2F), suggesting that performance of individuals with less flexible oscillators suffered more as detuning increased. Note that flexibility estimates quantified how much accuracy decreased as a function of trial-to-trial changes in stimulus rate (±𝚫IOI). Thus, these results show that oscillators that were robust to changes in stimulus rate were also less dependent on detuning to be able to synchronize across a wide range of stimulus rates. We are excited to be able to provide this extra validation of predictions made by entrainment models.

To revise the manuscript with the updated analysis on detuning:

We added the descriptions of the analyses to the Experiment 1 Methods section.

Calculation of detuning slopes and their averaging procedure are in Preferred rate estimates:

“In addition to estimating the preferred rate as stimulus rates with peak performance, we investigated whether accuracy increased as a function of detuning, namely, the difference between stimulus rate and preferred rate, as predicted by the entrainment models (Large, 1994; McAuley, 1995; Jones, 2018). We tested this prediction by assessing the slopes of mixed-effects logistic regression models, where accuracy was regressed on the IOI condition, separately for stimulus rates that were faster or slower than an individual’s preferred rate estimate. To do so, we first z-scored IOIs that were faster and slower than the participant’s preferred rate estimates, separately to render IOI scales comparable across participants. The detuning direction (i.e., whether stimulus IOI was faster or slower than the preferred rate estimate) was coded categorically. Accuracy (binary) was predicted by these variables (zscored IOI, detuning direction), and their interaction. The model was fitted separately to datasets from random-order and linear-order sessions, using the *fitglme* function in MATLAB. Fixed effects were z-scored IOI and detuning direction and random effect was their interaction. We expected a systematic increase in performance toward the preferred rate, which would result in a significant interaction between stimulus rate and detuning direction. To decompose the significant interaction and to visualize the effects of detuning, we fitted separate models to each participant’s single-session datasets, and obtained slopes from each direction condition, hereafter denoted as the ‘relative-detuning slope’. We treated relative-detuning slope as an index of the magnitude of relative detuning effects on accuracy. We then evaluated these models, using the *glmval* function in MATLAB to obtain *predicted* accuracy values for each participant and session. To visualize the relative-detuning curves, we averaged the predicted accuracies across participants within each session, separately for each direction condition (faster or slower than the preferred rate). To obtain a single value of relative-detuning magnitude for each participant, we averaged relative detuning slopes across direction conditions. However, since slopes from IOI > preferred rate conditions quantified an accuracy decrease as a function of detuning, we sign-flipped these slopes before averaging. The resulting average relative detuning slopes, obtained from each participant’s single-session datasets, quantified how much the accuracy increase towards preferred rate was dependent on, in other words, sensitive to, relative detuning.” (p. 7-8)

We added the information on the correlation analyses between average detuning slopes in Flexibility estimates.

“We further tested the relationship between the flexibility estimates (𝛽 from models where |𝚫IOI| or |+𝚫IOI| predicted accuracy) and average detuning slopes (see *Preferred rate estimates*) from random-order sessions. We predicted that flexible oscillators (larger 𝛽) would be less severely affected by detuning, and thus have smaller detuning slopes. Conversely, inflexible oscillators (smaller 𝛽) should have more difficulty in adapting to a large range of stimulus rates, and their adaptive abilities should be constrained around the preferred rate, as indexed by steeper relative detuning slopes.” (p. 8)

We provided the results in Experiment 1 Results section.

“Logistic models assessing a systematic increase in accuracy toward the preferred rate estimate in each session type revealed significant main effects of IOI (linear-order session: 𝛽 = 0.264, p < .001; random-order session: 𝛽 = 0.175, p < .001), and significant interactions between IOI and direction (linear-order session: 𝛽 = -0.444, p < .001; random-order session: 𝛽 = -0.364, p < .001), indicating that accuracy increased as fast rates slowed toward the preferred rate (positive slopes) and decreased again as slow rates slowed further past the preferred rate (negative slopes), regardless of the session type. Fig. 2B illustrates the preferred rate estimation method for an example participant’s dataset and shows the predicted accuracy values from models fitted to each participant’s single-session datasets. Note that the main effect and interaction were obtained from mixed effects models that included aggregated datasets from all participants, whereas the slopes quantifying the accuracy increase as a function of detuning (i.e., relative detuning slopes) were from models fitted to single-participant datasets.” (p. 9-10)

“We tested the relationship between the flexibility estimates and single-participant relative detuning slopes from random-order sessions (Fig. 2B). The results revealed negative correlations between the relative detuning slopes and flexibility estimates, both with 𝛽 (r(23) = 0.529, p = 0.007) from models where |-𝚫IOI| predicted accuracy (adapting to speeding-up trials), and 𝛽 (r(23) = -0.580, p = 0.002) from models where |+𝚫IOI| predicted accuracy (adapting to slowing-down trials). That is, the performance of individuals with less flexible oscillators suffered more as detuning increased. These results are shown in Fig. 2F.” (p. 10)

We modified Figure 2. In Figure 2B, there are now separate subfigures with the z-scored IOI faster (left) or slower (right) than the preferred rate predicting accuracy. We illustrated the correlations between average relative detuning slopes and flexibility estimates in Figure 2F.

**Author response image 1. sa3fig1:** Main findings of Experiment 1. (**A**) Left: Each circle represents a single participant’s preferred rate estimate from the random-order session (x axis) and linear-order session (y axis). The histograms along the top and right of the plot show the distributions of estimates for each session type. The dotted and dashed lines respectively represent 1:2 and 2:1 ratio between the axes, and the solid line represents one-to-one correspondence. Right: permutation test results. The distribution of summed residuals (distance of data points to the closest y=x, y=2*x and y=x/2 lines) of shuffled data over 1000 iterations, and the summed residual from original data (dashed line) that fell below .008 of the permutation distribution. (**B**) Top: Illustration of the preferred rate estimation method from an example participant’s linear-order session dataset. Estimates were the stimulus rates (IOI) where smoothed accuracy (orange line) was maximum (arrow). The dotted lines originating from the IOI axis delineate the stimulus rates that were faster (left, IOI < preferred rate) and slower (right, IOI > preferred rate) than the preferred rate estimate and expand those separate axes, the values of which were Z-scored for the relative-detuning analysis. Bottom: Predicted accuracy, calculated from single-participant models where accuracy in random-order (purple) and linear-order (orange) sessions was predicted by z-scored IOIs that were faster than a participant’s preferred rate estimate (left), and by those that were slower (right). Thin lines show predicted accuracy from single-participant models, solid lines show the averages across participants and the shaded areas represent standard error of the mean. Predicted accuracy is maximal at the preferred rate and decreases as a function of detuning. (**C**) Average accuracy from random-order (left, purple) and linear-order (right, orange) sessions. Each circle represents a participant’s average accuracy. (**D**) Flexibility estimates. Each circle represents an individuals’ slope (𝛽) obtained from logistic models, fitted separately to conditions where |𝚫IOI| (left, green) or |+𝚫IOI| (right blue) predicted accuracy, with greater values (arrow’s direction) indicating better oscillator flexibility. The means of the distributions of 𝛽 from both conditions were smaller than zero (dashed line), indicating a negative effect of between-trial absolute rate change on accuracy. (**E**) Participants’ average bias from |𝚫IOI| (green), and |+𝚫IOI| (blue) conditions in random-order (left) and linear-order (right) sessions. Negative bias indicates underestimation of the comparison intervals, positive bias indicates the opposite. Box plots in C-E show median (black vertical line), 25th and 75th percentiles (box edges) and extreme data points (whiskers). In C and E, empty circles show outlier values that remained after data cleaning procedures. F Correlations between participants’ average relative detuning slopes, indexing the steepness of the increase in accuracy towards the preferred rate estimate (from panel B), and flexibility estimates from |-𝚫IOI| (top, green), and |+𝚫IOI| (bottom, blue) conditions (from panel C). Solid black lines represent the best-fit line, dashed lines represent 95% confidence intervals.

We discussed the results in General Discussion and emphasized that only entrainment models, compared to timekeeper models, predict a relationship between detuning and accuracy that is amplified by oscillator’s inflexibility: “we observed systematic increases in task accuracy (Experiment 1) toward the best-performance rates (i.e., preferred rate estimates), with the steepness of this increase being closely related to the effects of rate change (i.e., oscillator flexibility). Two interdependent properties of an underlying system together modulating an individual’s timing responses show strong support for the entrainment approach” (p. 24)

“As a side note, colours blue/green have a different meaning in 2B than 2D and E, which might be confusing.”

Upon the reviewer’s recommendation, we changed the color scale across Figure 2, such that colors refer to the same set of conditions across all panels.

(2) Second, I don't understand the additional harmonic relationship analyses in the appendix, and I suspect other readers will not either. As with the previous point, it is not my view that the analyses are faulty or inadequate, it is rather that the lack of clarity makes it challenging to evaluate whether they support an entrainment model or not.

We decided to remove the analysis that was based on a circular approach, and we have clarified the analysis that was based on a modular approach by giving example cases:

“We first calculated how much the slower estimate (larger IOI value) diverts, proportionally from the faster estimate (smaller IOI value) or its multiples (i.e., harmonics) by normalizing the estimates from both sessions by the faster estimate. The outcome measure was the modulus of the slower, with respect to the faster estimate, divided by the faster estimate, described as mod(max(X), min(X))/min(X) where X = [session1_estimate session2_estimate]. An example case would be a preferred rate estimate of IOI = 603 ms from the linear-order session and an estimate of IOI = 295 ms from the random-order session. In this case, the slower estimate (603 ms) diverts from the *multiple* of the faster estimate (295*2 = 590 ms) by 13 ms, a proportional deviation of 4% of the faster estimate (295 ms). The outcome measure in this example is calculated as mod(603,295)/295 = 0.04.” (Supplementary Information, p. 2)

Crucially, the ability of oscillators to respond to harmonically-related stimulus rates is a main distinction between entrainment and interval (timekeeper) models. In the current study, we found that each participant’s best-performance rates, the preferred rate estimates, had harmonic relationships. The additional analyses further showed that these harmonic relationships were not due to chance. This finding speaks against the interval (timekeeper) approaches and is maximally compatible with the entrainment framework.

Here are a number of questions I would like to list to sketch my confusion:The authors write: "We first normalized each participant's estimates by rescaling the slower estimate with respect to the faster one and converting the values to radians". Does slower estimate mean: "task accuracy in those trials in which IOI was slower than a participant's preferred frequency"?

Preferred rate estimates were stimulus rates (IOI) with best performance, as described in Experiment 1 Methods section.

“We conceptualized individuals' preferred rates as the stimulus rates where durationdiscrimination accuracy was highest. To estimate preferred rate on an individual basis, we smoothed response accuracy across the stimulus-rate (IOI) dimension for each session type, using the smoothdata function in Matlab. Estimates of preferred rate were taken as the smoothed IOI that yielded maximum accuracy” (p. 7).

The estimation method and the resulting estimate for an example participant was provided in Figure 2B. The updated figure in the current revision has this illustration only for linear-order session.

“Estimates were the stimulus rates (IOI) where smoothed accuracy (orange line) was maximum (arrow)” (Figure caption, p. 9).

"We reasoned that values with integer-ratio relationships should correspond to the same phase on a unit circle". What is values here; IOI, or accuracy values for certain IOIs? And why should this correspond to the same phase?

We removed the analysis on integer-ratio relationships that was based on a circular approach that the reviewer is referring to here. We clarified the analysis that was based on a modular approach and avoided using the term ‘values’ without specifying what values corresponded to.

Des "integer-ratio relationships" have to do with the y=x, y=x*2 and y=x/2 relationships of the other analyses?

Integer-ratio relationships indeed refer to y=x, y=x*2 and y=x/2 relationships. For example, if a number y is double of another number x (y = x*2), these values have an integer-ratio relationship, since 2 is an integer. This holds true also for the case where y = x/2 since x = y*2.

Supplementary Figure S2c shows a distribution of median divergences resulting from the modular approach. The p-value is 0.004 but the dashed line appears to be at a much higher percentile of the distribution. I find this hard to understand.

We thank the reviewer for a detailed inspection of all figures and information in the manuscript. The reviewer’s comment led us to realize that this figure had an error. We updated the figure in Supplementary Information (Supplementary Figure S2).

**Reviewer #2 (Public Review):**

To get a better understanding of the mechanisms underlying the behavioral observations, it would have been useful to compare the observed pattern of results with simulations done with existing biophysical models. However, this point is addressed if the current study is read along with this other publication of the same research group: Kaya, E., & Henry, M. J. (2024, February 5). Modeling rhythm perception and temporal adaptation: top-down influences on a gradually decaying oscillator. https://doi.org/10.31234/osf.io/q9uvr

We agree with the reviewer that the mechanisms underlying behavioral responses can be better understood by modeling approaches. We thank the reviewer for acknowledging our computational modeling study that addressed this concern.

**Reviewer #2 (Recommendations):**

I very much appreciate the thorough work done by the authors in assessing all reviewers' concerns. In this new version they clearly state the assumptions to be tested by their experiments, added extra analyses further strengthening the conclusions and point the reader to a neurocomputational model compatible with the current observations.I only regret that the authors misunderstood the take home message of our Essay (Doelling & Assaneo 2021). Despite this being obviously out of the scope of the current work, I would like to take this opportunity to clarify this point. In that paper, we adopted a Stuart-Landau model not to determine how an oscillator should behave, but as an example to show that some behaviors usually used to prove or refute an underlying "oscillator like" mechanism can be falsified. We obviously acknowledge that some of the examples presented in that work are attainable by specific biophysical models, as explicitly stated in the essay: "There may well be certain conditions, equations, or parameters under which some of these commonly held beliefs are true. In that case, the authors who put forth these claims must clearly state what these conditions are to clarify exactly what hypotheses are being tested."This work did not mean to delineate what oscillator is (or in not), but to stress the importance of explicitly introducing biophysical models to be tested instead of relying on vague definitions sometimes reflecting the researchers' own beliefs. The take home message that we wanted to deliver to the reader appears explicitly in the last paragraph of that essay: "We believe that rather than concerning ourselves with supporting or refuting neural oscillators, a more useful framework would be to focus our attention on the specific neural dynamics we hope to explain and to develop candidate quantitative models that are constrained by these dynamics. Furthermore, such models should be able to predict future recordings or be falsified by them. That is to say that it should no longer be sufficient to claim that a particular mechanism is or is not an oscillator but instead to choose specific dynamical systems to test. In so doing, we expect to overcome our looping debate and to ultimately develop-by means of testing many model types in many different experimental conditions-a fundamental understanding of cognitive processes and the general organization of neural behavior."

We appreciate the reviewer’s clarification of the take-home message from Doelling and Assaneo (2021). We concur with the assertions made in this essay, particularly regarding the benefits of employing computational modeling approaches. Such methodologies provide a nuanced and wellstructured foundation for theoretical predictions, thereby minimizing the potential for reductionist interpretations of behavioral or neural data.

In addition, we would like to underscore the significance of delineating the level of analysis when investigating the mechanisms underlying behavioral or neural observations. The current study or Kaya & Henry (2024) involved no electrophysiological measures. Thus, we would argue that the appropriate level of analysis across our studies concerns the theoretical mechanisms rather than how these mechanisms are implemented on the neural (physical) level. In both studies, we aimed to explore or approximate the *theoretical* oscillator that guides dynamic attention rather than the neural dynamics underlying these theoretical processes. That is, theoretical (attentional) entrainment may not necessarily correspond to neural entrainment, and differentiating these levels could be informative about the parallels and differences between these levels.

References

Doelling, K. B., & Assaneo, M. F. (2021). Neural oscillations are a start toward understanding brain activity rather than the end. *PLoS Biol*, *19*(5), e3001234. https://doi.org/10.1371/journal.pbio.3001234 Jones, M. R. (2018). *Time will tell: A theory of dynamic attending*. Oxford University Press.Kaya, E., & Henry, M. J. (2024). Modeling rhythm perception and temporal adaptation: top-down influences on a gradually decaying oscillator. *PsyArxiv*.10.31234/osf.io/q9uvr.Large, E. W. (1994). *Dynamic representation of musical structure*. The Ohio State University.McAuley, J. D. (1995). *Perception of time as phase: Toward an adaptive-oscillator model of rhythmic pattern processing* Indiana University Bloomington.Pikovsky, A., Rosenblum, M., & Kurths, J. (2003). *Synchronization: A Universal Concept in Nonlinear Sciences*. Cambridge University Press.